# *Wolbachia* endosymbionts manipulate the self-renewal and differentiation of germline stem cells to reinforce fertility of their fruit fly host

Shelbi L. Russell[1]*, Jennie Ruelas Castillo[2], William T. Sullivan[3]

**1** Department of Biomolecular Engineering, University of California Santa Cruz, Santa Cruz, California, United States of America, **2** Division of Infectious Diseases, Department of Medicine, The Johns Hopkins Hospital, Baltimore, Maryland, United States of America, **3** Department of Molecular, Cell, and Developmental Biology, University of California Santa Cruz, Santa Cruz, California, United States of America

* shelbilrussell@gmail.com

**Data Availability Statement:** All relevant data are contained within the paper, Supporting Information files, NCBI BioProject number PRJNA1007602 or on Dryad: https://doi.org/10.7291/D1DT2C.

## Abstract

The alphaproteobacterium *Wolbachia pipientis* infects arthropod and nematode species worldwide, making it a key target for host biological control. *Wolbachia*-driven host reproductive manipulations, such as cytoplasmic incompatibility (CI), are credited for catapulting these intracellular bacteria to high frequencies in host populations. Positive, perhaps mutualistic, reproductive manipulations also increase infection frequencies, but are not well understood. Here, we identify molecular and cellular mechanisms by which *Wolbachia* influences the molecularly distinct processes of germline stem cell (GSC) self-renewal and differentiation. We demonstrate that *w*Mel infection rescues the fertility of flies lacking the translational regulator *mei-P26* and is sufficient to sustain infertile homozygous *mei-P26*-knockdown stocks indefinitely. Cytology revealed that *w*Mel mitigates the impact of *mei-P26* loss through restoring proper pMad, Bam, Sxl, and Orb expression. In Oregon R files with wild-type fertility, *w*Mel infection elevates lifetime egg hatch rates. Exploring these phenotypes through dual-RNAseq quantification of eukaryotic and bacterial transcripts revealed that *w*Mel infection rescues and offsets many gene expression changes induced by *mei-P26* loss at the mRNA level. Overall, we show that *w*Mel infection beneficially reinforces host fertility at mRNA, protein, and phenotypic levels, and these mechanisms may promote the emergence of mutualism and the breakdown of host reproductive manipulations.

## Introduction

Endosymbiotic bacteria have evolved diverse strategies for infecting and manipulating host populations [1,2], which are now being leveraged for biological control applications [3]. Many of these bacteria reside within host cells and navigate female host development to colonize offspring, thus linking their fitness to that of their hosts through vertical transmission [4]. Inherited endosymbionts with reproductive manipulator capabilities go a step further by altering

**Funding:** This work was supported by the UC Santa Cruz Chancellor's Postdoctoral Fellowship and the NIH (R00GM135583 to SLR; R35GM139595 to WTS). The funders had no role in study design, data collection and analysis, decision to publish, or preparation of the manuscript.

**Competing interests:** The authors have declared that no competing interests exist.

**Abbreviations:** BDSC, Bloomington Drosophila Stock Center; BMP, bone morphogenic protein; CB, cystoblast; CC, cystocyte; CI, cytoplasmic incompatibility; CTCF, corrected total cell fluorescence; ERR, estrogen-related receptor; GSC, germline stem cell; NDJ, non-disjunction; OreR, Oregon R; pHH3, phospho-Histone H3; PI, propidium iodide; RISC, RNA-induced silencing complex; RNP, ribonucleoprotein; TA, transit-amplifying; TomO, toxic manipulator of oogenesis; UTR, untranslated region.

host development in ways that rapidly increase the frequency of infected, reproductive females in the host population. Reproductive manipulator strategies include cytoplasmic incompatibility (CI), where infected sperm require rescue by infected eggs, male-killing, feminization, and parthenogenesis [5,6]. These manipulations can be highly effective at driving bacterial symbionts into host populations, regardless of costs to the host individual or population [7].

Despite the success of parasitic reproductive manipulation, natural selection favors symbionts that increase the fertility of infected mothers, even if these variants reduce the efficacy of the parasitic mechanisms that initially drove the infection to high frequency [8]. In associations between arthropods and strains of the alphaproteobacterium *Wolbachia pipientis*, this scenario may be common. For example, measured fecundity in populations of *Drosophila simulans* infected with the wRi strain of *Wolbachia* swung from −20% to +10% across a 20-year span following wRi's CI-mediated sweep across California in the 1980s [9]. This transition from fitness cost to benefit coincided with weakening of CI strength over the same time frame [10]. Fertility-enhancing mechanisms may be at work in other strains of *Wolbachia* that have reached high infection frequencies in their native hosts, yet do not currently exhibit evidence of parasitic reproductive manipulation [11]. Importantly, these fitness benefits may have also played a role in the early stages of population infection, when infected hosts are at too low of frequencies for CI to be effective [12].

Currently, the *w*Mel strain of *Wolbachia* and its encoded CI mechanism are successfully being used to biologically control non-native hosts [3]. In *Aedes aegypti* mosquitoes, CI causes nearly 100% mortality of offspring born to uninfected mothers [13]. However, in its native host, the fruit fly *Drosophila melanogaster*, *w*Mel exhibits CI that rarely exceeds 50% mortality and is extremely sensitive to paternal age [14–16], as well as grandmother age because titer increases with age [17,18]. Despite weak CI in its native host, *w*Mel is found at moderate to high infection frequencies in populations worldwide [19,20]. Other data suggest that these frequencies may be explained by some emergent beneficial function that increases host fitness [21–25]. The molecular basis for these beneficial functions could be related to the loss of CI efficacy in *D. melanogaster*. Given that *w*Mel's use in non-native hosts relies on strong and efficient CI, it is essential that we learn the basis for its beneficial functions that could ultimately undermine CI function.

Host germline stem cell (GSC) maintenance and differentiation pathways are powerful targets for *Wolbachia*-mediated reproductive manipulation. *Wolbachia* strains have strong affinities for host germline tissues [26,27], positioning them at the right place to manipulate and enhance host fertility. In the strains that form obligate associations with *Brugia* filarial nematodes [28] and *Asobara* wasps [29], *Wolbachia* is required in the germline to prevent premature differentiation and achieve successful oogenesis (reviewed in [30]). In the facultative *w*Mel-*D. melanogaster* association, the *w*Mel strain can partially rescue select loss of function alleles of the essential germline maintenance and differentiation genes *sex lethal (sxl)* and *bag-of-marbles (bam)* in female flies [31–33]. It is known that *w*Mel encodes its own factor, toxic manipulator of oogenesis (TomO), that partially recapitulates Sxl function in the GSC through derepression and overexpression of the translational repressor Nanos (Nos) [34,35]. However, Nos expression is negatively correlated with Bam expression in the early germarium [36]. Therefore, Bam's function in cyst patterning and differentiation in *w*Mel-infected mutant flies cannot be explained by shared mechanisms with *sxl* rescue or TomO's known functions.

In a previous screen, our lab identified the essential fertility gene *meiotic-P26* (*mei-P26*) as a host factor that influences *w*Mel infection intensity in *D. melanogaster* cell culture, potentially through modulating protein ubiquitination [37]. Mei-P26 is a Trim-NHL protein that confers a wide range of functions through its multiple domains: its protein-binding NHL and B-box domains allow it to act as an adapter for multiple translational repressor complexes (e.g.,

Sxl, Nanos (Nos), Argonaute-1 (AGO1), see S1 to S2A Figs and S1 to S2 Tables). Its E3-ubiquitin ligase domain likely enables it to modulate proteolysis [38]. Given *mei-P26's* in vitro role in infection [37] and its in vivo role in GSC maintenance [38], differentiation [39,40], and meiosis [40,41], we selected it as a candidate gene for modulating *w*Mel–host interactions.

We report that *D. melanogaster* flies infected with the *w*Mel strain of *Wolbachia* partially rescue *mei-P26* mutations and exhibit reinforced fertility. In flies homozygous for hypomorphic *mei-P26*, infection with *w*Mel elevates fertility in both males and females to a level sufficient to maintain a stable stock, whereas the uninfected stock is unsustainable. Infection rescues GSC maintenance and cyst differentiation by mitigating the downstream effects of perturbed *mei-P26* function on other protein and mRNA expression. Specifically, *w*Mel infection restores a wild-type-like expression profile for phosphorylated Mothers against decapentaplegic (pMad), Sxl, Bam, and Oo18 RNA-binding (Orb) proteins, as well as *tumorous testis* (*tut*) and *benign gonadal cell neoplasm* (*bgcn*) mRNAs. We find evidence of *w*Mel's beneficial reproductive manipulator abilities in Oregon R (OreR) wild-type flies, illustrating how bacterially mediated developmental resilience may be selected for in nature. These results are essential to understanding how *w*Mel reaches high frequencies in natural *D. melanogaster* populations and these beneficial functions may be able to be harnessed for biological control applications.

## Results

Here, we explore the effects of *w*Mel *Wolbachia* infection on *D. melanogaster's* essential fertility gene *mei-P26* through dosage knockdown with an RNAi construct, disrupted function with a hypomorphic allele, *mei-P26[1]*, and full knockdown with a null allele, *mei-P26[mfs1]*. Using fly fecundity assays, immunocytochemistry, and dual-RNAseq of both host and bacterial transcripts, we show that *w*Mel can compensate for the loss of *mei-P26* to significantly rescue host fertility.

### Infection with wMel rescues female *D. melanogaster* fertility in mei-P26 deficient flies

Infection with *w*Mel significantly rescued fertility defects induced by the loss of *mei-P26* function relative to the OreR wild-type strain, as measured by offspring produced per female *D. melanogaster* per day (Figs 1A–1E and S3; $p< = 2.2e-16$ to $2.9e-2$, Wilcoxon rank sum test; see S2–S6 Figs). Offspring production requires successful maintenance of the GSC, differentiation of a GSC daughter cell into a fully developed and fertilized egg, embryogenesis, and hatching into a first-instar larva. Breaking offspring production down into egg lay and egg hatch components (Fig 1B and 1C) revealed that as allelic strength increased, from *nos*-driven RNAi to hypomorphic and null alleles, fertility impacts shifted from those that impacted differentiation/development (egg hatch) to those that also impacted egg production (egg lay).

Infection with *w*Mel mediated a significant degree of fertility rescue for all allelic strengths and nos:GAL4>UAS-driven RNAi knockdowns, suggesting a robust bacterial rescue mechanism that compensates for loss of protein dosage and function. Rescue was more than complete for *nos*-driven *mei-P26* RNAi depletion (from 26% deficient offspring production in uninfected flies): *w*Mel-infected *mei-P26* RNAi females produced 44% more offspring than infected OreR wild-type females (37 versus 25 larvae/female/day, $p< = 1.15E-02$ Wilcoxon rank sum test, S4 Table and Fig 1A and 1E) due to a higher rate of egg production ($p< = 6.99E-05$ Wilcoxon rank sum test, S5 Table and Fig 1B and 1E), opposed to a higher hatch rate (Fig 1C and 1E and S6 Table). This may suggest a synergistic function between *w*Mel and low-dose *mei-P26* in the GSC. In uninfected and *w*Mel-infected *mei-P26[1]* hypomorphic females, offspring production decreased 94% and 73%, respectively, compared to OreR wild-type of the

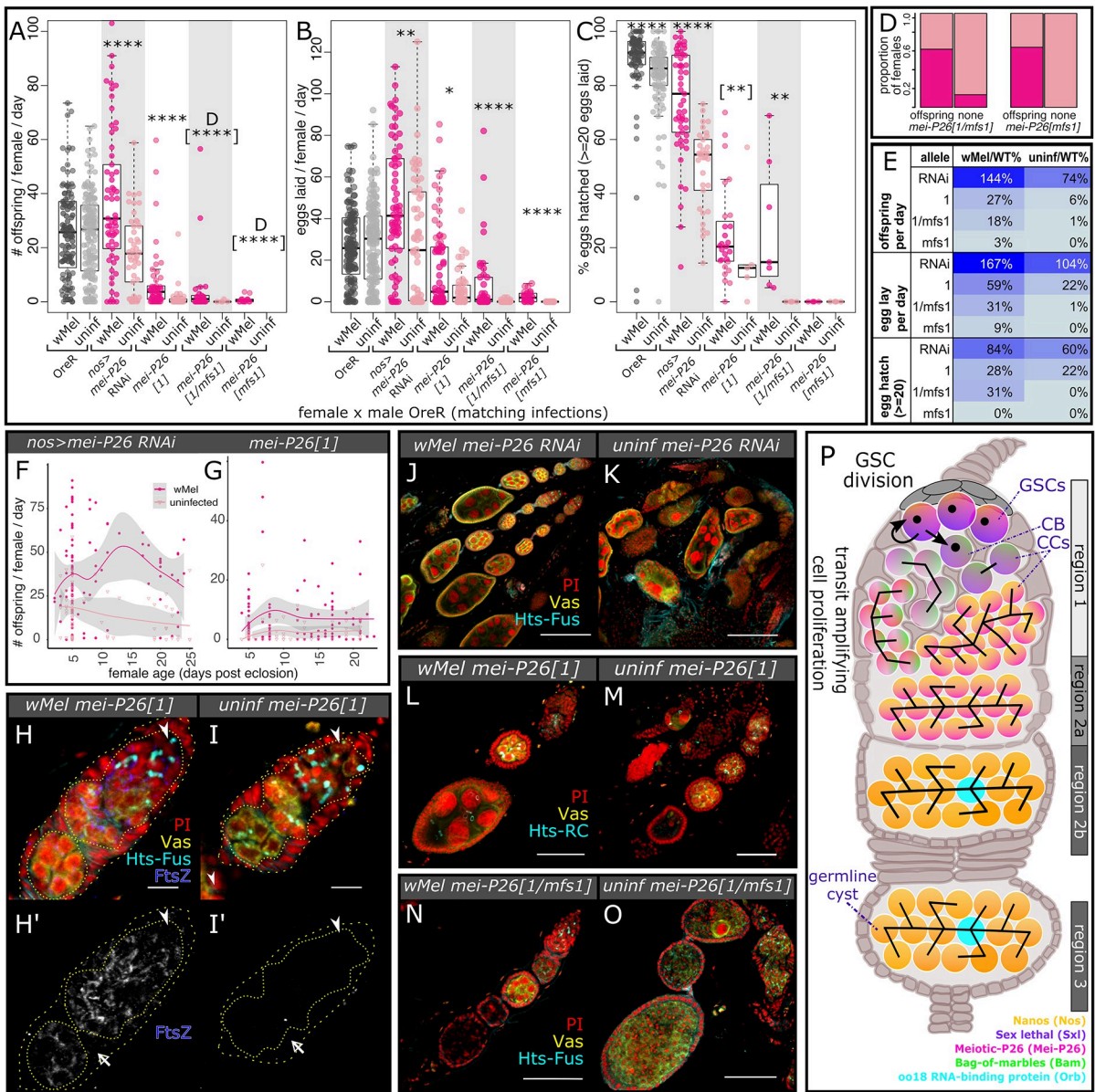

**Fig 1. *w*Mel infection rescues the loss of essential host fertility gene *mei-P26* in females (and males, see S4E–S4G Fig).** (A–C) Beeswarm boxplots of (A) overall offspring production, broken into (B) eggs laid and (C) eggs hatched, from single female RNAi, hypomorphic, and null *mei-P26* knockdown female flies crossed to single OreR males of the same infection state and age (3–7 days old). The null allele *mei-P26[mfs1]* reduced egg lay rates in both mutant heterozygotes and homozygotes below an average of 20 for both infected and uninfected females, necessitating an analysis of females with and without offspring (see D). Wilcoxon rank sum test *$p <$ = 0.05, **$<$ = 0.01, ***$<$ = 0.001, ****$<$ = 1e-4. Fisher's exact test [**]$p <$ = 0.01, [****]$<$ = 1e-4. (D–E) (D) Barplots of the proportion of female samples of the indicated genotypes with and without offspring, infected and uninfected with *Wolbachia*. (E) Table listing the percentage of wild-type fecundity demonstrated by *w*Mel-infected (left) or uninfected (right) *mei-P26* mutants in order of increasing severity, from top to bottom (see also S4–S6 Tables). Cell values from low to high proportions are colored from light to dark blue, respectively. (F, G) Knockdown *mei-P26 D. melanogaster* fecundity versus female age, fit with a local polynomial regression (dark gray bounds = 95% confidence intervals). Infection with *w*Mel increases offspring production over the first 25 days of the (F) *mei-P26* RNAi and (G) *mei-P26[1]* hypomorphic female lifespan. (H–O) Confocal mean projections of *D. melanogaster* ovarioles and germaria (3–7 days old). (H–I') Intracellular *w*Mel FtsZ localizes to germline (yellow, Vas+, inner dotted outline) and somatic (Vas-, space between inner dotted and outer dashed outlines) cells at high titers, with low background in uninfected flies. Bacteria are continuously located in the *mei-P26[1]* germline, starting in the GSC (arrowhead near Hts-bound spectrosome). Somatic cell regions are indicated with empty arrowheads. (J–O) Cytology reveals that *w*Mel-infection rescues (J, K) *mei-P26* RNAi, (L, M) *mei-P26[1]* (see also S5 Fig), and (N, O) *mei-P26[1/mfs1]*-induced oogenesis defects. Fluorescence channels were sampled serially and overlaid as indicated on each set of images as follows: cyan = anti-Hu Li Tai Shao (Hts) ring canal isoform (-RC) or fusome isoform (-FUS), yellow = anti-Vasa (Vas), blue = *w*Mel anti-Filamenting temperature-sensitive mutant Z (FtsZ), and red = PI DNA staining. Scale bars: H–I' = 10 μm, L–O = 50 μm, J, K = 100 μm. (P) Diagram of a wild-type *D. melanogaster* germarium, highlighting key

events and cell types by region as in [42]. Somatic cells in beige, GSC-derived cells are brightly colored gradients leading to differentiated cyst cells in yellow, with specified oocytes in blue. The black structure that originates in the GSC is the spectrosome, which becomes a branching fusome as the CB is formed, moves away from the niche, and divides into CCs. The data underlying this figure can be found in S3 Table and on Dryad at doi.org/10.7291/D1DT2C. CB, cystoblast; CC, cystocyte; GSC, germline stem cell; OreR, Oregon R; PI, propidium iodide.

matching infection status ($p<$ = 2.26E-14 and 3.45E-12 Wilcoxon rank sum test, respectively, S4 Table and Fig 1A and 1E). Comparing uninfected and *w*Mel-infected female *mei-P26[1]* fecundity revealed that *w*Mel infection increases the number of offspring produced per day (1.6 versus 7 larvae/female/day, $p<$ = 9.37E-05 Wilcoxon rank sum test, S4 Table and Fig 1A) through increasing both the rate of egg lay ($p<$ = 2.82E-02 Wilcoxon rank sum test, S5 Table and Fig 1B) and egg hatch ($p<$ = 2.55E-03 Wilcoxon rank sum test, S6 Table and Fig 1C). In uninfected and *w*Mel-infected *mei-P26[1]/mei-P26[mfs1]* trans-heterozygous females, off-spring production decreased 99% and 82%, respectively, relative to OreR wild-type ($p<$ = 3.40E-04 and 5.01E-10 Wilcoxon rank sum test, S4 Table and Fig 1A and 1D). Comparing uninfected and *w*Mel-infected trans-heterozygous flies showed that infection elevates offspring production (0.2 versus 4.6 larvae/female/day, $p<$ = 1.03E-05 Fisher's exact test, S4 Table and Fig 1A) through increasing the rate females lay eggs ($p<$ = 4.71E-05 Wilcoxon rank sum test; Fig 1B) and the rate those eggs hatch ($p<$ = 4.20E-03 Wilcoxon rank sum test; S6 Table and Fig 1C). In females homozygous for the most severe allele, *mei-P26[mfs1]*, offspring production decreased 100% and 97% in uninfected and *w*Mel-infected flies relative to wild-type, respectively ($p<$ = 5.14E-06 Fisher's exact test, S4 Table and Fig 1A). Offspring production was marginally rescued in *mei-P25[mfs1]* flies by *w*Mel infection (0 (uninfected) versus 0.76 (infected) larvae/female/day, $p<$ = 5.14E-06 Fisher's exact test, S4 Table, Fig 1A), in part due to infected flies laying eggs at a higher rate than uninfected flies ($p<$ = 3.04E-05 Wilcoxon rank sum test, Fig 1B). However, no female laid 20 or more eggs in one day, precluding an estimation hatch rate rescue (Fig 1C). See S2–S6 Tables and S3 Fig for a full description of the fecundity assays and data.

*w*Mel-mediated *mei-P26* rescue is robust and sufficient to rescue this gene in a stable stock. Infection with *w*Mel elevates the number of offspring produced per day across the fly lifespan in *mei-P26[1]* hypomorphic and *nos*-driven RNAi-knockdown females ($p<$ = 1.5e-11 to 3.4e-2 Kolmogorov–Smirnov test, S7 Table and Fig 1F and 1G). However, the underlying number of eggs laid per day and the hatch rate do vary considerably with age for both infection states and genotypes (S4A–S4D Fig). The *w*Mel rescue mechanism is not specific to females, as the weaker impacts of *mei-P26* loss on male fertility are mitigated by *w*Mel-infection (S4E–S4G Fig; $p<$ = 6.1e-6 to 2.9e-2 Wilcoxon rank sum test, S3–S5 Tables). Thus, we were able to establish a homozygous stock of *mei-P26[1]* flies infected with *w*Mel *Wolbachia*. In contrast, the uninfected stock only lasted a few generations without balancer chromosomes (S4H and S4I Fig). Given the severe, yet significantly rescuable nature of the *mei-P26[1]* allele (Figs 1A–1C, S2B and S2D), we proceeded with this genotype for many of our subsequent assays for specific immunocytological rescue phenotypes.

## Female germline morphology is rescued by wMel infection in mei-P26-deficient flies

In the *D. melanogaster* germarium, *w*Mel is continuously present at high titers in both germline and somatic cells (Fig 1H–1H', compared to control in Fig 1I–1I'), consistent with previous reports [26,27,43]. The bacteria localize more strongly to the germline-derived cells than the somatic cells (co-localization of FtsZ and Vas in Fig 1H–1H'). Intracellular *w*Mel can be identified in the GSC, the cystoblast (CB), the cystocytes (CCs), and the developed cyst. This

positioning puts *w*Mel in all of the critical cell types and stages that *mei-P26* is active, enabling the bacterium to compensate for *mei-P26's* developmental functions in GSC maintenance and differentiation.

Comparing the cytology of *mei-P26* knockdown oocytes infected and uninfected with *Wolbachia* confirmed that *w*Mel-infected ovarioles exhibit far fewer developmental defects than uninfected, more closely resembling OreR wild-type (Figs 1J–1O and S5 versus Fig 1P; Vasa (Vas) = germline [44], Hu Li Tai Shao (Hts) = cytoskeletal spectrosome/fusome [45], propidium iodide (PI) = DNA). Specifically, *w*Mel-infected *mei-P26*-knockdown ovarioles exhibited more normally formed cysts, consisting of somatically derived follicle cells surrounding germline-derived nurse cells and an oocyte, than uninfected ovarioles. Aberrant phenotypes included cysts lacking the follicle cell exterior (Fig 1K and 1M), the germline-derived interior (Fig 1M and 1N), or a normal number of differentiated nurse cells (Fig 1O).

These comparative data for a range of *mei-P26* alleles, RNAi-knockdowns, and host ages indicate that *w*Mel rescues *mei-P26's* developmental functions in early host oogenesis. In contrast, *w*Mel does not rescue *mei-P26's* role in meiosis. Segregation defects were not rescued by *w*Mel infection, as indicated by elevated X-chromosome nondisjunction (NDJ) rates in both infected and uninfected *mei-P26[1]* homozygous hypomorphs (NDJ = 7.7% and 5.6% by experiment, respectively, S6A and S6B Fig). In the following sections, we analyze *w*Mel's ability to rescue *mei-P26* function at each of the critical time points in early oogenesis.

## Host GSCs are maintained at higher rates in wMel-infected than uninfected mei-P26-deficient flies

GSCs were quantified by immunofluorescence staining of the essential GSC markers pMad and Hts in "young" 4- to 7-day-old (Fig 2) and "aged" 10- to 13-day-old females (S7A–S7C Fig). Positive pMad staining indicates the cell is responding to quiescent signals from the surrounding somatic GSC niche cells. Positive Hts staining requires localization to a single punctate spectrosome, opposed to a branching fusome [46].

Staining revealed an increase in the average number of GSCs per *mei-P26[1]* germarium with *w*Mel infection, relative to uninfected germaria in young, 4- to 7-day-old flies (infected/hypomorph: 1.0 versus uninfected/hypomorph: 0.58, *p*< = 2.9e-4 Wilcoxon rank sum test, S8 Table), but GSC maintenance did not reach OreR wild-type rates (infected/WT: 2.3 and uninfected/WT: 1.9, *p*< = 1.2e-4 to 2.8e-4 Wilcoxon rank sum test, Fig 2A–2E and S8 Table). *w*Mel-infected *mei-P26[1]* germaria also had more cells manifesting other GSC properties, such as a large cytoplasm and physical attachment to the cap cells of the somatic niche [46], than uninfected germaria. In contrast to the *mei-P26[1]* allele, RNAi knockdown of *mei-P26* did not significantly affect the numbers of GSCs per germarium (infected/RNAi: 2.1 and uninfected/RNAi: 1.9, S7A–S7C Fig and S8 Table), suggesting that the modest reductions in *mei-P26* dosage have a stronger impact on differentiation than GSC maintenance. OreR wild-type germaria did not exhibit different numbers of GSCs due to *w*Mel infection (Fig 2C–2E and S8 Table).

The number of GSCs per germarium in *w*Mel-infected *mei-P26[1]* females converges on OreR wild-type values in aged 10- to 13-day-old germaria (S7C Fig). As wild-type flies age, the number of GSCs per germarium declines naturally [47], even in *w*Mel-infected flies (aged/infected/WT: 1.6 versus young/infected/WT: 2.3, *p*< = 0.019 Wilcoxon rank sum test, Figs 2C–2E and S7C and S8 Table). Both infected and uninfected *mei-P26[1]* females appear to retain some of their GSCs as they age (aged/infected/hypomorph: 1.5 and aged/uninfected/hypomorph: 0.94, *p*< = 0.013–0.014 Wilcoxon rank sum test, S7C Fig and S8 Table). While OreR wild-type infected and uninfected flies lose from 3% to 31% of their GSCs in the first 2

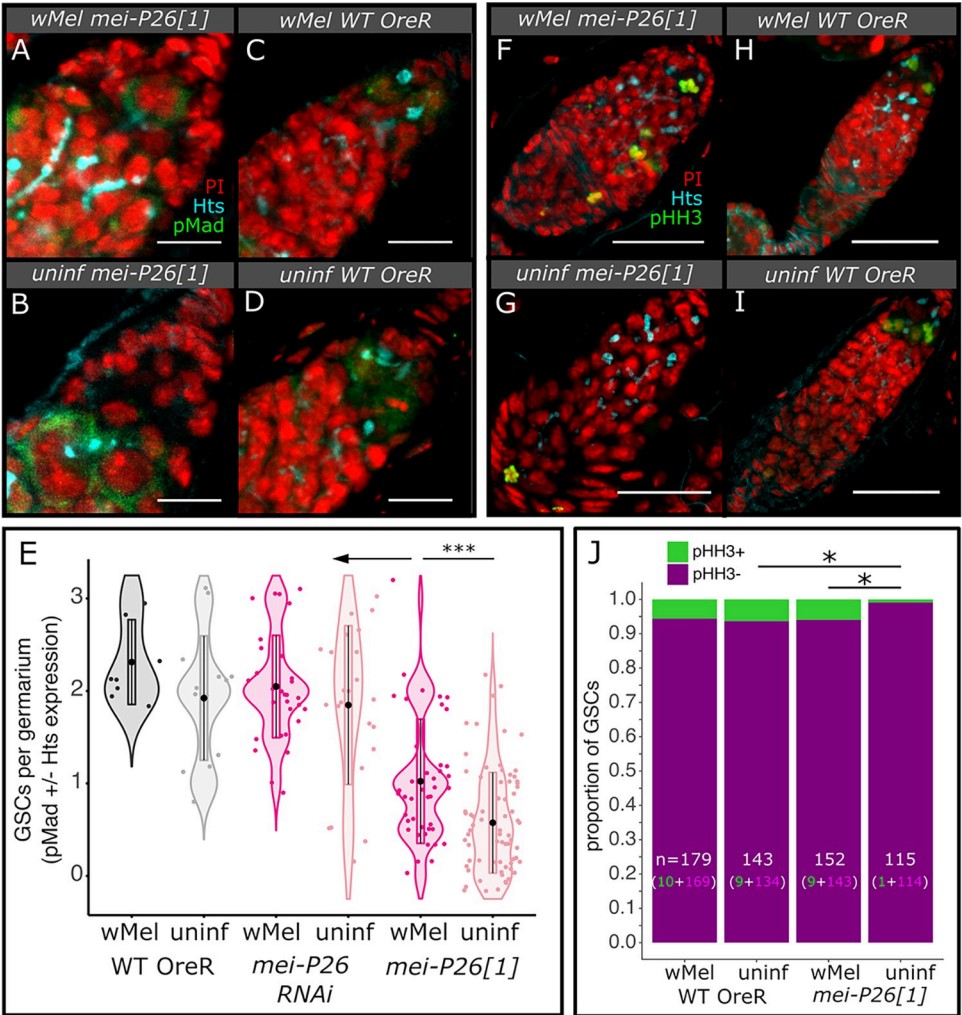

**Fig 2. *w*Mel infection rescues *mei-P26*-induced defects in female GSC maintenance.** (A–E) Infection with *w*Mel confers higher numbers of GSCs per germarium in *mei-P26[1]* germaria than in uninfected flies. (A–D) Confocal mean projections of *D. melanogaster* germaria stained with antibodies against Hts and pMad. (E) Violin plots of the number of GSCs per germarium. As fully functional GSCs express pMad and have Hts-labeled spectrosomes, each was weighted by half, allowing for partial scores. Wilcoxon rank sum test * = $p<0.05$, ** = 0.01, *** = 0.001, **** = 1e-4. (F–J) Mitotic GSCs detected by pHH3 expression revealed that *w*Mel restored mitosis in hypomorphic *mei-P26* GSCs relative to uninfected GSCs. (F–I) Confocal mean projections of *D. melanogaster* germaria stained with antibodies against Hts and pHH3. (J) Stacked bar chart of the proportion of GSCs found expressing pHH3 in female germaria. Fisher's exact test * = $p < 0.05$. Fluorescence channels were sampled serially and overlaid as indicated on each set of figs as follows: cyan = anti-Hts, yellow = anti-Vas, red = PI, green = anti-pMad in (A–D) and anti-pHH3 in (F–I). Scale bars A–D = 10 μm and F–I = 25 μm. The data underlying this figure can be found on Dryad at doi.org/10.7291/D1DT2C. GSC, germline stem cell; pHH3, phospho-Histone H3.

weeks, infected and uninfected *mei-P26[1]* flies increase their GSC abundance by 43% and 63% in this time (S7C Fig), potentially through recruiting differentiated GSCs [48]. Taking both age-dependent trajectories in GSC retention into account, by the time *w*Mel-infected *mei-P26[1]* germaria are 10 to 13 days old, they contain the same average number of GSCs per germarium as wild-type flies and significantly more GSCs than uninfected *mei-P26[1]* flies ($p< = 0.015$ Wilcoxon rank sum test, S7C Fig and S8 Table). Therefore, *Wolbachia* enhances stem cell maintenance in both young and old *mei-P26[1]* females.

Infected *mei-P26[1]* GSCs are functional, as indicated by their ability to divide by mitosis (Fig 2F–2J and S9 Table). We identified nuclei in mitosis by positive anti-phospho-histone H3 serine 10 (pHH3) staining, a specific marker of Cdk1 activation and mitotic entry [49]. Significantly fewer GSCs were in mitosis in uninfected *mei-P26[1]* germaria than infected *mei-P26[1]* and uninfected OreR wild-type germaria (0.09% versus 6%, $p$ = 4.7e-2 and 4.6e-2 Fisher's exact test, Fig 2F–2J and S9 Table). There was no difference in the frequency of GSCs in mitosis between uninfected and infected wild-type germaria (5% to 6%, Fig 2H–2J and S9 Table). Mitotic cystoblasts and cystocytes were only significantly enriched in infected *mei-P26[1]* germaria compared to infected wild-type germaria ($p$< = 1.3e-2 Wilcoxon rank sum test, S7D Fig and S10 Table), despite over-replication during transit-amplifying (TA) mitosis being a common phenotype in uninfected *mei-P26[1]* germaria (discussed below and [50]).

## Regulation of host Sxl expression is rescued in wMel-infected germaria

Proper Sxl expression is required for GSC maintenance and differentiation [51]. High levels of Sxl in the GSC maintains stem cell quiescence [52]. When the GSC divides and the CB moves away from the niche, Sxl cooperates with Bam and Mei-P26 to bind to the *nos* 3′ untranslated region (UTR) and down-regulate Nos protein levels to promote differentiation [50,53].

We quantified Sxl localization patterns in whole ovaries stained with anti-Sxl antibodies by confocal microscopy of whole oocytes, revealing that Sxl dysregulation due to *mei-P26* loss is mitigated by wMel infection (Fig 3). Less Sxl is expressed in uninfected *mei-P26[1]* germarium region 1 and more Sxl is expressed across regions 2a and 2b, relative to OreR wild-type ($p$< = 5.7e-10 to 5.3e-5 Wilcoxon rank sum test, Fig 3A–3B' versus 3F–3H' and 3I, S7E Fig, and S11 Table). Infection with wMel increased Sxl expression in germarium region 1, had no impact on region 2a, and suppressed expression in region 2b relative to uninfected *mei-P26[1]*, replicating an expression pattern similar to that seen in wild-type germaria ($p$< = 5.7e-10 to 8.4e-3 Wilcoxon rank sum test, Figs 3A–3B' versus 3C–3E' and 3I and S7E and S11 Table). Interestingly, wMel infection may impact Sxl expression in OreR germaria, as the protein is significantly elevated in region 2a of wMel-infected relative to uninfected germaria (Fig 3I). Thus, Sxl dysregulation due to *mei-P26* loss can be partially rescued by the presence of wMel, suggesting that wMel's mechanism for *mei-P26* rescue may underlie how Nos regulation in wMel-infected Sxl mutants [31,34].

## Bam expression is partially restored in wMel-infected mei-P26 mutant germaria

In wild-type female flies, Bam expression begins immediately after the cystoblast daughter cell moves away from its undifferentiated sister, which remains bound to the GSC niche ([50,54] and illustrated in Fig 1P). Bam binds to the fusome and stabilizes CyclinA to promote TA mitosis, producing 16 cyst cells from a single CB [55]. Immediately following these 4 mitoses, Bam expression is down-regulated in wild-type ovaries to limit germline cysts to 16 germline-derived cells.

Through anti-Bam immunofluorescence confocal imaging and quantification, we determined that wMel-infection mitigates aspects of Bam dysregulation in *mei-P26[1]* germaria (Fig 4). Loss of Mei-P26 deregulates and extends Bam expression past germarium region 1 (Fig 4A–4B" versus 4C–4F" and 4G; S7F Fig and S12 Table), producing excess nurse-like cells [50]. Infection with wMel enables Bam up-regulation at the CB-to-16-cell stage in *mei-P26[1]* germaria relative to uninfected germaria ($p$< = 0.016, Fig 4G). Bam may also be down-regulated in stage 2a and 2b wMel-infected *mei-P26[1]* germaria because uninfected germaria express significantly more Bam than wild-type ($p$< = 0.019 Wilcoxon rank sum test, Fig 4G

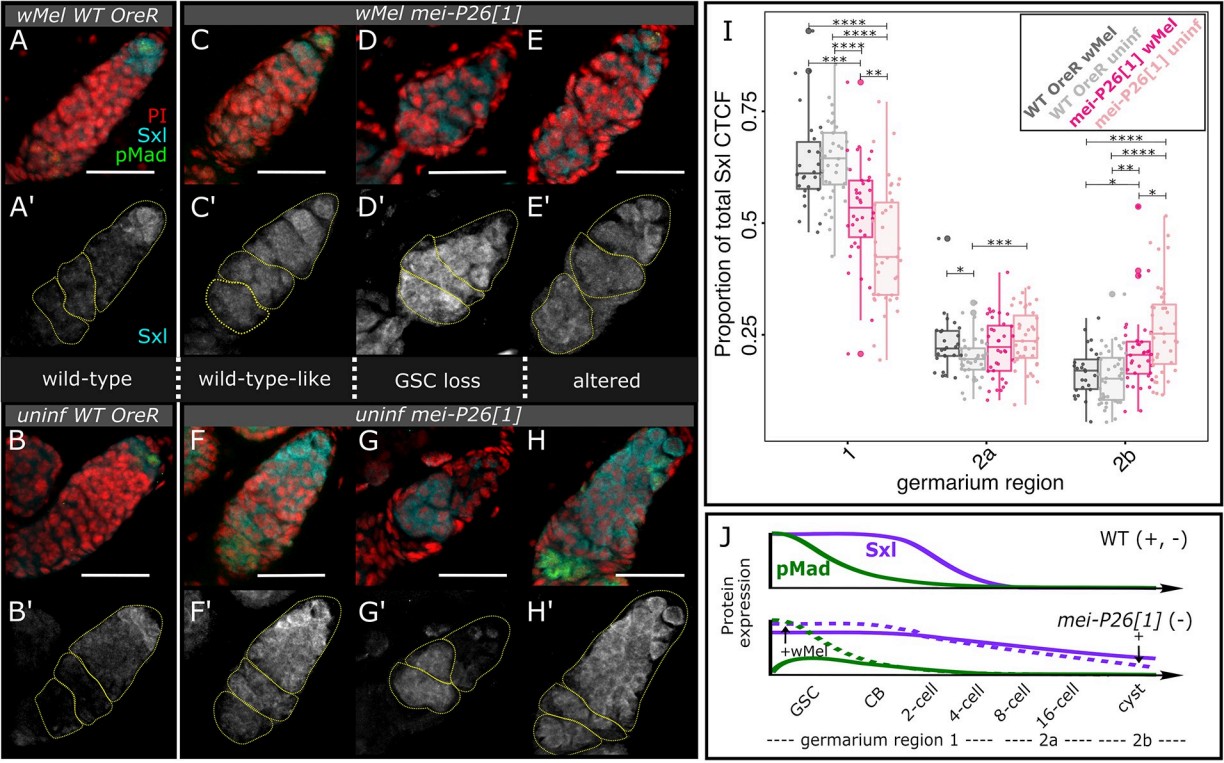

**Fig 3. Infection with *w*Mel mitigates the consequences of dysfunctional *mei-P26* on Sxl expression in the germarium.** (A–H') Confocal mean projections of *D. melanogaster* germaria stained with antibodies against Sxl and pMad. Three phenotypes for each *mei-P26[1]* infection state are shown, (left) one as similar to wild-type as possible, (middle) one representative of GSC loss, pMad-negative, and (right) one representative of a developmentally altered phenotype retaining pMad-positive GSCs. We sampled the fluorescence channels serially and overlaid them as indicated on each set of images: cyan = anti-Sxl, red = PI, green = anti-pMad. Scale bars = 25 μm. (I) Bar-scatter plots of relative Sxl fluorescence expression levels across the germarium, by region (yellow outlines over Sxl-channel images in A'–H'). Colors as labeled in Figs 1A–1D and 2E, from left to right: dark gray = *w*Mel-infected OreR, light gray = uninfected OreR, dark pink = *w*Mel-infected *mei-P26[1]*, light pink = uninfected *mei-P26[1]*. Wilcoxon rank sum test * = $p < 0.05$, ** = 0.01, *** = 0.001, **** = 1e-4. (J) Model of *w*Mel Sxl expression rescue in *mei-P26[1]* germaria. Green dashed line represents *w*Mel's up-regulation of pMad expression in the GSC (Fig 2A–2E). Purple dashed line represents *w*Mel's up-regulation of Sxl in the GSC/CB and down-regulation of Sxl by germarium region 2b. Arrows with + symbols indicate *w*Mel's action on gene expression. The data underlying this figure can be found on Dryad at doi.org/10.7291/D1DT2C. CB, cystoblast; GSC, germline stem cell; OreR, Oregon R; PI, propidium iodide.

and S12 Table), whereas wMel-infected germaria do not (Fig 4G and S12 Table). Unfortunately, variance in *mei-P26[1]* Bam fluorescence intensity among samples is too large to resolve whether uninfected and infected germaria differ in Bam expression after the 16-cell stage with this dataset.

Infection with *w*Mel alters Bam expression in the GSC and CB to achieve an expression profile similar to and more extreme than wild-type germaria, respectively (Fig 4H and 4I). Both infected and uninfected *mei-P26[1]* germaria exhibit lower Bam expression than wild-type in the GSC niche (Fig 4G). This contrasts with expectations from the literature: bone morphogenic protein (BMP) signaling from the somatic niche induces pMad expression in the GSC, which directly represses Bam transcription, preventing differentiation [56]. Loss of *mei-P26* function in the GSC should derepress Brat, resulting in pMad repression and inappropriate Bam expression [38]. However, we see lower Bam expression in both *w*Mel-infected and uninfected *mei-P26[1]* GSCs relative to OreR wild-type ($p < = 1.3$e-6 to 1.4e-3 Wilcoxon rank sum test, Fig 4G and S13 Table). The elevated relative Bam/pMad expression ratio in uninfected *mei-P26[1]* GSCs ($p < = 2.5$e-4 to 4.0e-4 Wilcoxon rank sum test, Fig 4H and 4I and

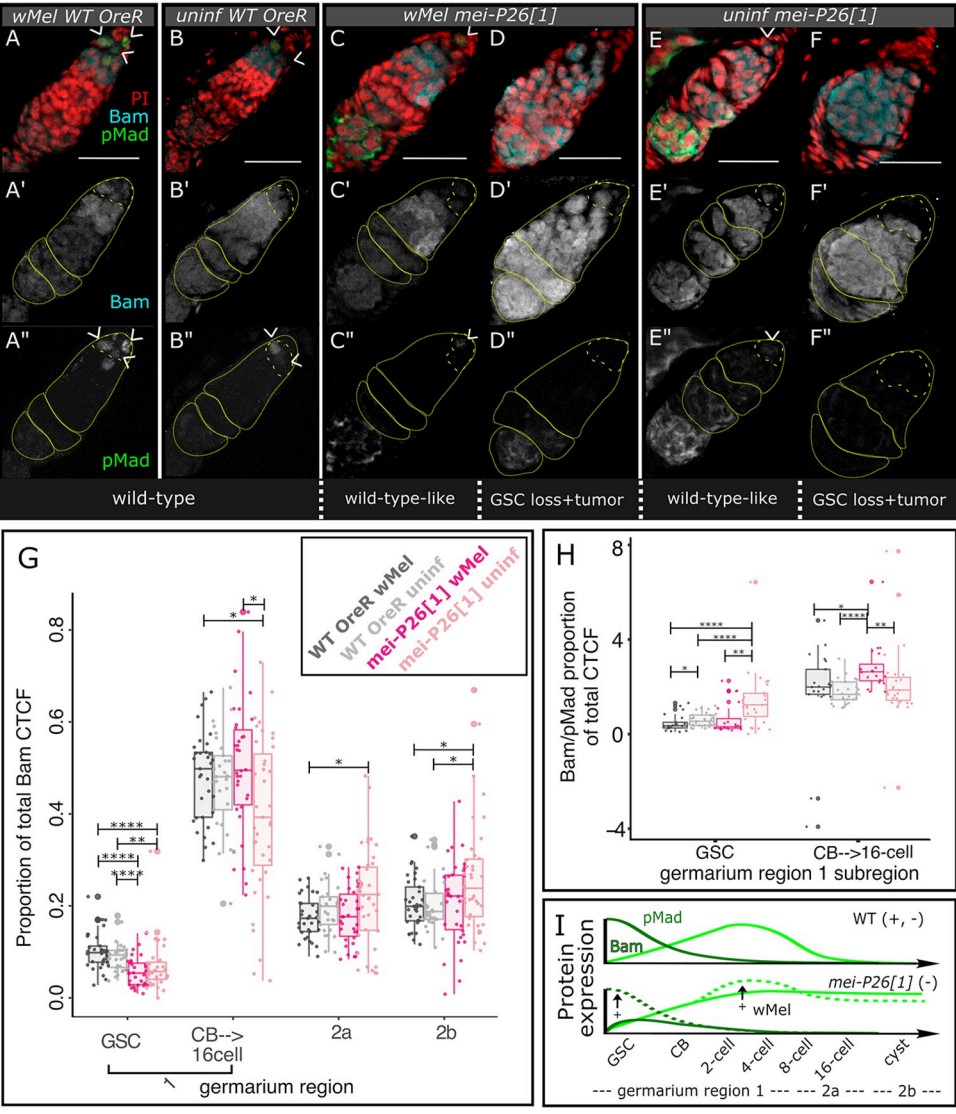

**Fig 4. Infection with *w*Mel mitigates the consequences of dysfunctional *mei-P26* on *Bam* expression in the germarium.** (A–F") Confocal mean projections of *D. melanogaster* germaria stained with antibodies against Bam and pMad. Two phenotypes for each *mei-P26[1]* infection state are shown, (left) one as similar to wild-type as possible and (right) one representative of developmentally altered phenotypes. (G) Bar-scatter plots of relative Bam fluorescence expression levels across the germarium, by region (yellow outlines over Bam-channel images in A'–F'). We subdivided region 1 into GSC and CB/cystocyte subsections because Bam expression ramps up in the CB. (H) Bar-scatter plot of the ratio of relative Bam to pMad fluorescence. As pMad and Bam expression are mutually exclusive in the GSC and CB/cystocytes, respectively, OreR wild-type values are near zero. Values significantly higher than this reflect dysregulation of the stem cell state. (I) Model of *w*Mel rescue of Bam expression in *mei-P26[1]* germaria. Colors as labeled in Figs 1A–1C, 2E and 3I, from left to right: dark gray = *w*Mel-infected OreR, light gray = uninfected OreR, dark pink = *w*Mel-infected *mei-P26[1]*, light pink = uninfected *mei-P26[1]*. Wilcoxon rank sum test * = $p<0.05$, ** = 0.01, **** = 1e-4. Scale bars = 25 μm. The data underlying this figure can be found on Dryad at doi.org/10.7291/D1DT2C. CB, cystoblast; GSC, germline stem cell; OreR, Oregon R.

S13 Table) may explain this departure from expectations. Although GSC Bam expression is lower than in wild-type, Bam expression is clearly dysregulated and up-regulated relative to pMad expression, as expected for germline cells that have lost their stem cell identity [57,58]. Normalizing against pMad expression also reinforces the finding that *w*Mel elevates Bam

expression even higher than wild-type levels in *mei-P26[1]* hypomorphs at the CB-to-16-cell stage ($p <$ = 1.2e-6 to 2.0e-2 Wilcoxon rank sum test, Fig 4H and 4I and S13 Table).

## *w*Mel rescues normal germline cyst and oocyte development impaired by *mei-P26* knockdown

Mei-P26-deficient flies over-proliferate nurse cells and partially differentiate GSCs to produce tumorous germline cysts ([38,39] and Fig 5A–5D versus 5E–5H, Fig 5I and S14 Table). In RNAi knockdown ovaries, the formation of tumorous germline cysts is significantly mitigated by *w*Mel infection (35.3% versus 16.3% of cysts with more than 15 nurse cells, respectively; *p* = 4.5e-3 Fisher's exact test, Fig 5E and 5F versus 5G and 5H and S14 Table). In contrast, germline cyst tumors found in *mei-P26[1]* allele germaria were not rescued by *w*Mel infection, despite the bacterium's ability to rescue the rate at which these eggs hatch into progeny (Fig 5I versus Fig 1A and 1C), but consistent with their dysregulation of Bam expression in germarium regions 2a and 2b (Fig 4G). The stronger nature of the *mei-P26[1]* allele relative to the nos-driven RNAi knockdown likely underlies the difference in the number of tumorous cysts. Lack of tumor rescue indicates that either tumorous cysts produce normal eggs at some rate or *w*Mel infection rescues tumors later in oogenesis, perhaps through some somatic mechanism [59–62].

Overexpression of *mei-P26* produces tumors in both uninfected and infected ovaries; however, infected ovaries exhibit significantly more tumors than uninfected ovaries (*p* = 1.6e-3 Fisher's exact test, Fig 5I and S14 Table). Finding that an excess of *mei-P26* recapitulates the tumorous phenotype induced by a lack of *mei-P26* indicates that this gene exerts a concentration-dependent dominant negative effect on its own function. Given that infection with *w*Mel makes this dominant negative phenotype more severe, we hypothesize that *w*Mel may make a factor that acts to enhance Mei-P26 function, potentially through a direct interaction or redundant function.

Infection with *w*Mel increases the rate of oocyte-specific Orb localization in *mei-P26[1]* germline cysts (Figs 5J–5R and S7G and S15 Table), suggesting that *w*Mel rescues functional cyst formation. Oocyte specification in region 2a of the germarium (see diagram in Fig 1P) is inhibited by the loss of *mei-P26* through the derepression of *orb* translation [38]. Mei-P26 binds to the *orb* 3′ UTR and prevents its translation. Following cyst patterning and development, one of the 16 germline-derived cells is designated as the oocyte through specific oo18 RNA-binding protein (Orb) expression [63]. In region 2a, the germline cyst cell fated to become the oocyte activates *orb* translation [38]. Loss of *mei-P26* results in *orb* derepression and nonspecific expression [38]. Infection with *w*Mel restores oocyte-specific Orb expression in *mei-P26[1]* cysts relative to uninfected cysts (*p* = 5.9e-9 Fisher's exact test, Fig 5L–5N' versus 5O–5Q' and S15 Table). We observed oocyte-specific Orb staining 93% to 94% of the time in wild-type cysts, whereas *mei-P26[1]* reduced this frequency to 25%, and *w*Mel infection recovered *mei-P26[1]* cysts to 61% (counts in S15 Table). Oocyte-specific expression of Orb in *w*Mel-infected cysts is robust, even in those that are developmentally aberrant (e.g., the double oocyte in Fig 5M–5M'). These results suggest that *w*Mel rescues oocyte-specific Orb translation enhancing or duplicating Mei-P26's function in *orb* translational repression.

## OreR wild-type host fertility is beneficially impacted by wMel infection

Consistent with *w*Mel's ability to rescue the partial to complete loss of essential germline maintenance genes encompassing a range of functions, we discovered that this intracellular symbiont can reinforce fertility in *D. melanogaster* stocks displaying full wild-type fertility (Figs 1C and S8A–S8C and S4–S6 Tables). We analyzed egg lay, egg hatch, and overall offspring

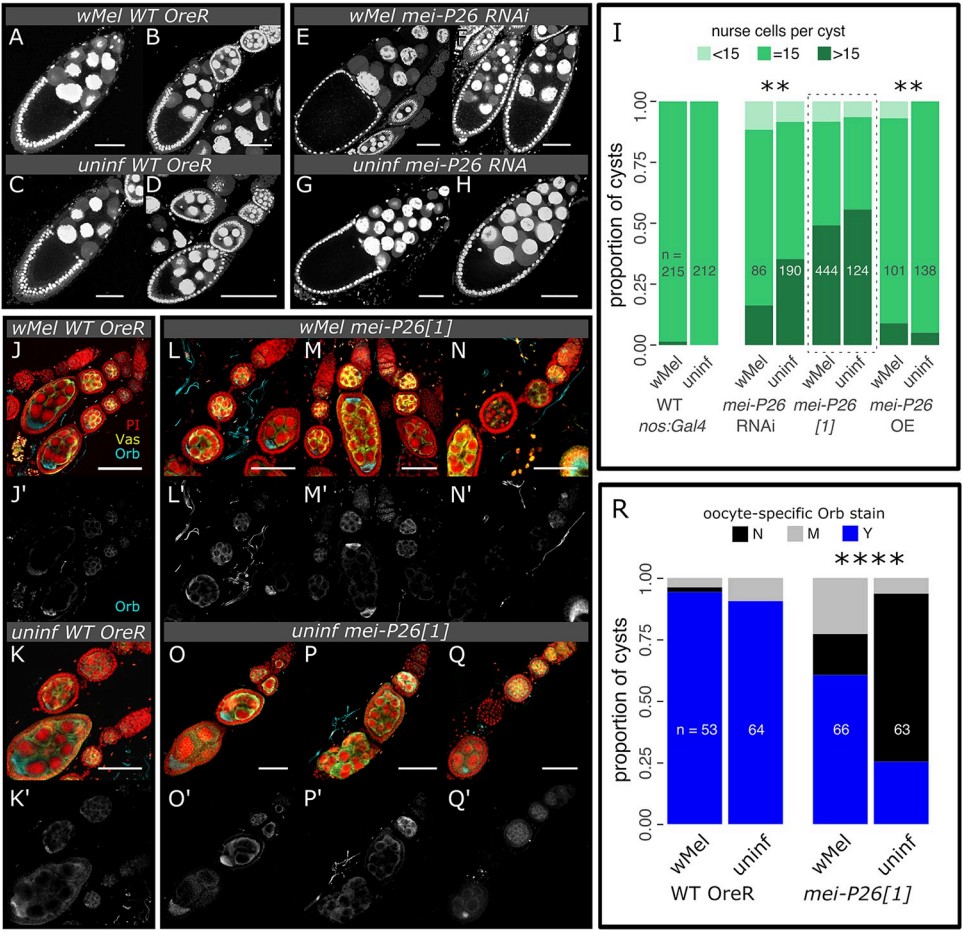

**Fig 5. *w*Mel infection rescues cyst tumors and restores oocyte specification in *mei-P26* mutants.** (A–I) Loss and overexpression of Mei-P26 induces the formation of tumorous cysts, which contain more than 15 nurse cells. *w*Mel infection mitigates *nos>mei-P26* RNAi-knockdown and exacerbates *nos>mei-P26* OE phenotypes. (A–H) Confocal max projections of approximately 25 μm-thick sections of *D. melanogaster* oocyte cysts stained with PI. (I) Stacked bar chart of the proportion of germline cysts containing an abnormal number of nurse cells (less than or greater than 15 nurse cells). (J–R) Hypomorphic *mei-P26[1]* cysts infected with *w*Mel significantly restored Orb translational regulation relative to uninfected cysts. Oocyte specification occurs in regions 2b-3 of the germarium (see diagram in Fig 1P), when Orb expression becomes restricted to the nascent oocyte. (J–Q') Confocal mean projections of *D. melanogaster* germaria stained with antibodies against Orb and Vas. (R) Stacked bar chart of the proportion of germline cysts containing oocyte-specific Orb staining. Sample sizes are written on the bar charts. N = no, M = maybe, and Y = yes. Fisher's exact test ** = $p<$1e-2, **** = $p<$1e-4. Scale bars = 50 μm. The data underlying this figure can be found on Dryad at doi.org/10.7291/D1DT2C. OE, overexpression; PI, propidium iodide.

production rates independently to detect *w*Mel effects on multiple components of fertility. Overall offspring production per female per day was elevated in *w*Mel-infected nos:Gal4/CyO females relative to uninfected females by 49% (27 (uninfected) versus 40 (infected) offspring/female/day, $p< = 2.2$e-3 Wilcoxon rank sum test, S8A Fig and S4 Table). Infection with *w*Mel increased the number of eggs laid per female per day for the nos:Gal4/CyO balancer stock by 49% (42 versus 28 eggs/female/day, $p< = 2.2$e-3 Wilcoxon rank sum test, S8B Fig and S5 Table), but did not have any detectable effect on egg lay rates in other genotypes with wild-type fertility. Following embryogenesis, infection with *w*Mel elevated the rate that OreR and nos:Gal4/CyO wild-type eggs hatch into L1 stage larvae by 5.2% and 2.5%, respectively (88% and 91% (infected) versus 83% and 89% (uninfected) of eggs hatched, $p< = 1$e-4 to 2.1e-3

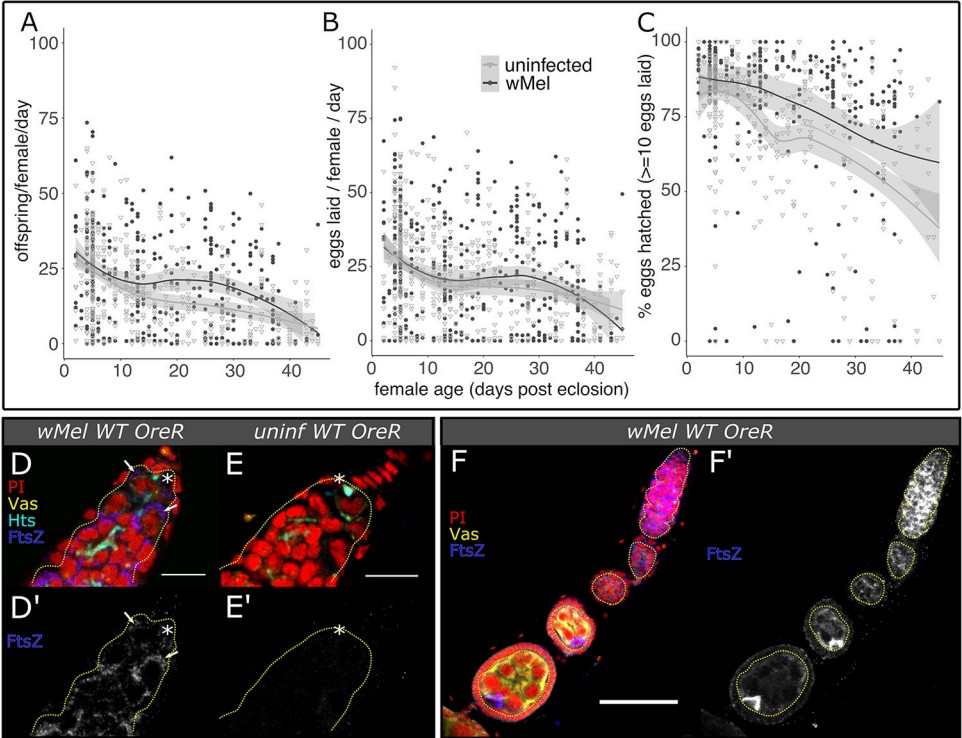

**Fig 6. The *w*Mel strain of *Wolbachia* reinforces OreR wild-type fertility.** (A–C) *D. melanogaster* OreR fecundity versus female age, fit with a local polynomial regression (dark gray bounds = 95% confidence intervals). Infection with *w*Mel may elevate (A) offspring and (B) egg offspring production at time points during the host lifespan through continuously elevating (C) hatch rate. (D–F') Confocal mean projections of *D. melanogaster* (D–D') infected and (E–E') uninfected germaria and (F–F') an infected ovariole. Intracellular *w*Mel FtsZ localizes to germline (Vas+, outlined in dashed yellow) and somatic (Vas-) cells at high titers in infected OreR wild-type flies, with low background in uninfected flies. Bacteria in the GSC (* near the Hts-bound spectrosome) are indicated with arrows. Fluorescence channels were sampled serially and overlaid as indicated on each set of images as follows: cyan = anti-Hts, yellow = anti-Vas, blue = anti-FtsZ, and red = PI. Scale bars F–G = 10 μm, H = 50 μm. The data underlying this figure can be found in S3 Table and on Dryad at doi.org/10.7291/D1DT2C. GSC, germline stem cell; OreR, Oregon R; PI, propidium iodide.

Wilcoxon rank sum test, Figs 1C and S8C and S6 Table). These results suggest that beneficial impacts of *w*Mel infection are evident, yet variable among *D. melanogaster* stocks with wild-type fertility.

We measured OreR fertility across its lifetime (approximately 50 days) and found that *w*Mel may produce higher lifetime fecundity relative to uninfected flies (Fig 6A–6C and S7 Table). OreR females were aged as indicated along the x-axis of Fig 6A–6C and mated to 3 to 7 days old males of matching infection status. *w*Mel-infection maintains elevated OreR egg hatch rates as flies age ($p < $ = 7.1e-9 Kolmogorov–Smirnov test, Fig 6C and S7 Table), while maintaining similar levels of egg production as uninfected flies (Fig 6B and S7 Table). This culminates in age ranges in which infected flies may produce an excess of offspring relative to uninfected flies (see gap between the regression confidence intervals when flies were approximately 20 to 30 days old, Fig 6A). The efficiency gains imparted on hosts by infection-induced elevated hatch rates could potentially accumulate for higher lifetime fitness, as resources are more efficiently converted to offspring.

Immunostaining of anti-FtsZ confirms that *w*Mel exhibits high titers in the OreR wild-type germarium and concentrates in the oocyte in early cysts (Fig 6D–6F'). The intracellular bacteria are continuously located in the germline, starting in the GSC and proceeding through the

end of embryogenesis and fertilization. Thus, *w*Mel are located in the right place at the right time to reinforce female host fertility through interactions with the GSC and oocyte cyst, as well as to correct for perturbations caused by male CI-related alterations [64].

The *w*Mel strain of *Wolbachia* in its native *D. melanogaster* host exhibits weak CI that decreases with age (S9A–S9D Fig), suggesting low maintenance of CI mechanisms [8]. Mating *w*Mel-infected OreR males to infected (a "rescue cross") and uninfected OreR virgin females (a "CI cross") revealed that *w*Mel reduces uninfected egg hatch by 26% when males are 0 to 1 day old ($p<$ = 4.6e-4 Wilcoxon rank sum test, S9A Fig and S6 Table), but this moderate effect is eliminated by the time males are 5 days old, on average (S9C Fig and S6 Table). Only young males significantly impact offspring production ($p<$ = 1.7E-2 Wilcoxon rank sum test, S9B and S9D Fig and S4 Table). In contrast, the *Wolbachia* wRi strain that naturally infects *Drosophila simulans* (the Riv84 line) and induces strong CI [10,65], reduces uninfected egg hatch by 95% when males are 0 to 1 day old ($p<$ = 5.6e-8 Wilcoxon rank sum test, S9E Fig and S6 Table) and only loses some of this efficacy by 5 days (75% hatch reduction, $p<$ = 3.6e-4, Wilcoxon rank sum test; S9G Fig, S6 Table). This significantly impacts overall offspring production ($p<$ = 6.5E-7 and 5.0E-2 Wilcoxon rank sum test, S9F and S9H Fig and S4 Table). Paternal grandmothers of CI offspring were 3 to 7 days old (i.e., relatively young), contributing to weak CI in the *D. melanogaster* crosses [17]. Overall, these results suggest that *w*Mel's beneficial reproductive manipulations may affect its fitness more than its negative reproductive manipulations in nature.

## *w*Mel-mediated *mei-P26[1]* rescue may be enabled through rescuing and perturbing host transcription

Exploring *w*Mel's effect on host gene expression in OreR and *mei-P26[1]* ovaries with dual-eukaryotic and bacterial RNA sequencing (dual-RNAseq) revealed that infection significantly alters host gene expression in a variety of ways (Fig 7A–7C and S16–S19 Tables), which indicate a few potential mechanisms for *w*Mel-mediated fertility rescue. Of the 35,344 transcripts in the *D. melanogaster* genome, 10,720 were expressed at high enough levels in at least 5 of 6 samples of each genotype+infection group to be included in the analysis. Of the 1,286 genes in the *w*Mel genome, 663 were expressed in *D. melanogaster* ovaries (Fig 7D and S20 Table).

Both uninfected and *w*Mel-infected hypomorphic *mei-P26[1]* ovaries exhibited a 5-fold depletion in *mei-P26* expression relative to OreR ovaries (S2B and S2D Fig) and significant dysregulation of 1,044 other genes due to the *mei-P26[1]* allele after FDR correction (Wald test FDR-adjusted *p*-value (padj)$<$ = 0.05, Figs 7A, 7E and S10A and S17 Table). The changes in gene expression due to the loss and dysfunction of *mei-P26* far outnumber and outpace the changes in gene expression due to infection, in either *D. melanogaster* genotype, and highlight how important *mei-P26* is as a global regulator of gene expression. This number is well in excess of the 366 genes thought to be involved in GSC self-renewal [66], underscoring how many processes *mei-P26* is involved in, from germline differentiation [50] to somatic development [61]. Across biological process, cell component, and molecular function GO categories, loss of *mei-P26* impacted processes involving chromatin, recombination, protein–protein interactions, and muscle cell differentiation (S10B–S10G Fig), supporting *mei-P26's* known role in genetic repression, differentiation, and meiosis [38,40,50].

Infection with *w*Mel significantly alters the expression of hundreds of *D. melanogaster* genes to either restore or compensate for *mei-P26*-regulated gene expression. We tested for genes significantly associated with infection state in DESeq2 under the full model: ~genotype + infection + infection*genotype, revealing 422 significantly differentially expressed genes (Wald test padj$<$ = 0.05, Fig 7B and S18 Table). Testing for an effect of the interaction between

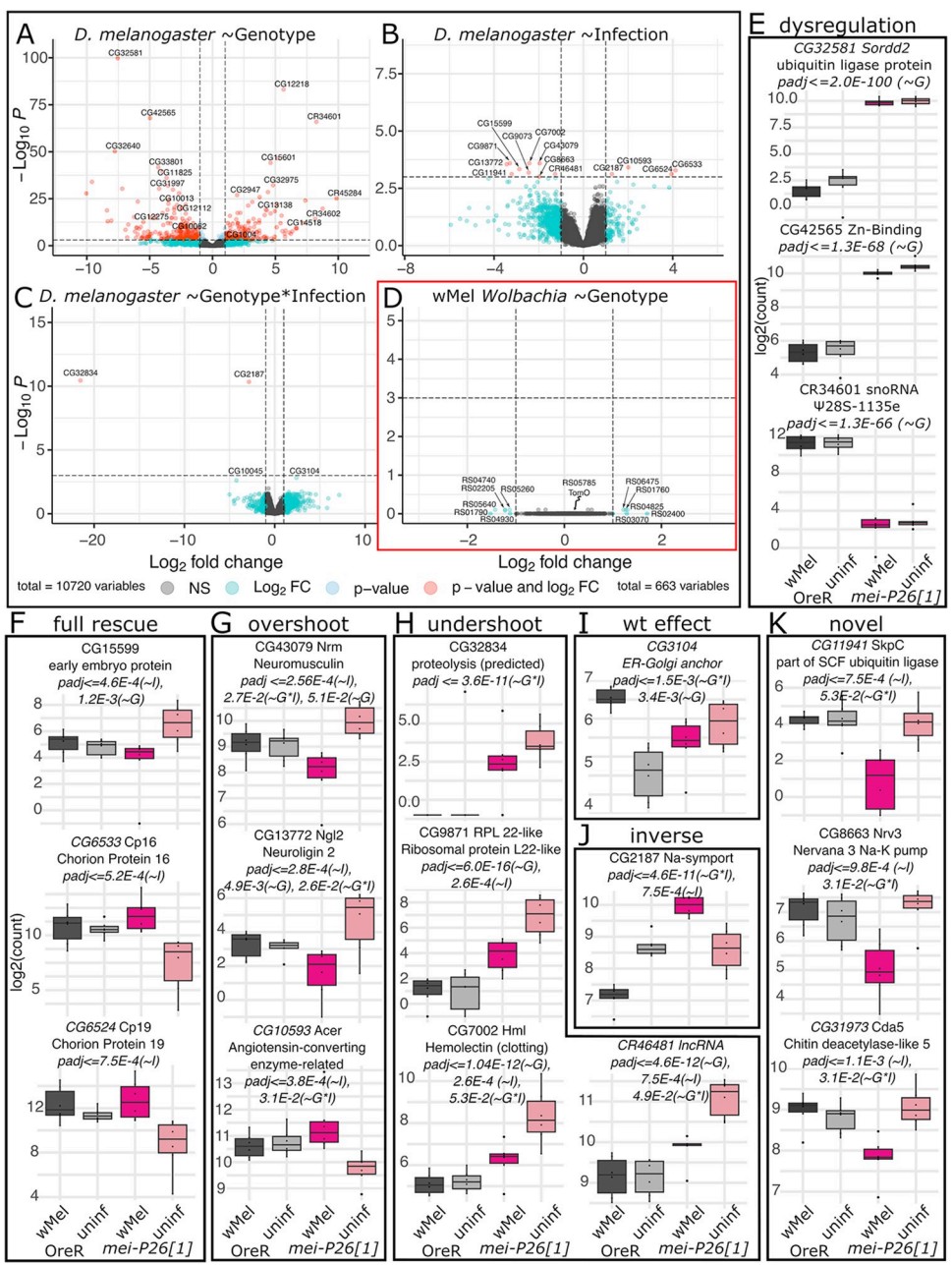

**Fig 7. Differential expression analysis of *w*Mel and *D. melanogaster* genes from host ovaries reveal that *mei-P26* knockdown and infection influence global expression patterns.** (A–D) Volcano plots of log2-fold change in gene expression versus log10-FDR adjusted *p*-value (padj) for the test performed in each panel. (E–K) Normalized transcript count plots for (E) the 3 top genotype-associated (~*G*) Wald Test hits by padj, and (F–K) *D. melanogaster* genes significantly associated (padj< = 1e-3) with infection (~*I*) or jointly, genotype and infection (~*G*I*), organized by the differential expression pattern. See S10–S15 Figs for other significant and insignificant count plots. In all barplots, dark gray = *w*Mel-infected OreR, light gray = uninfected OreR, dark pink = *w*Mel-infected *mei-P26[1]*, light pink = uninfected *mei-P26[1]*. The data underlying this figure can be found at NCBI, under BioProject number PRJNA1007602. OreR, Oregon R.

infection and genotype yielded another 20 significantly differentially expressed genes (Wald test padj< = 0.05; Fig 7C and S19 Table). The most significantly differentially expressed genes are featured in Fig 7F–7K and other significant genes are plotted in Figs 8, S11 and S12.

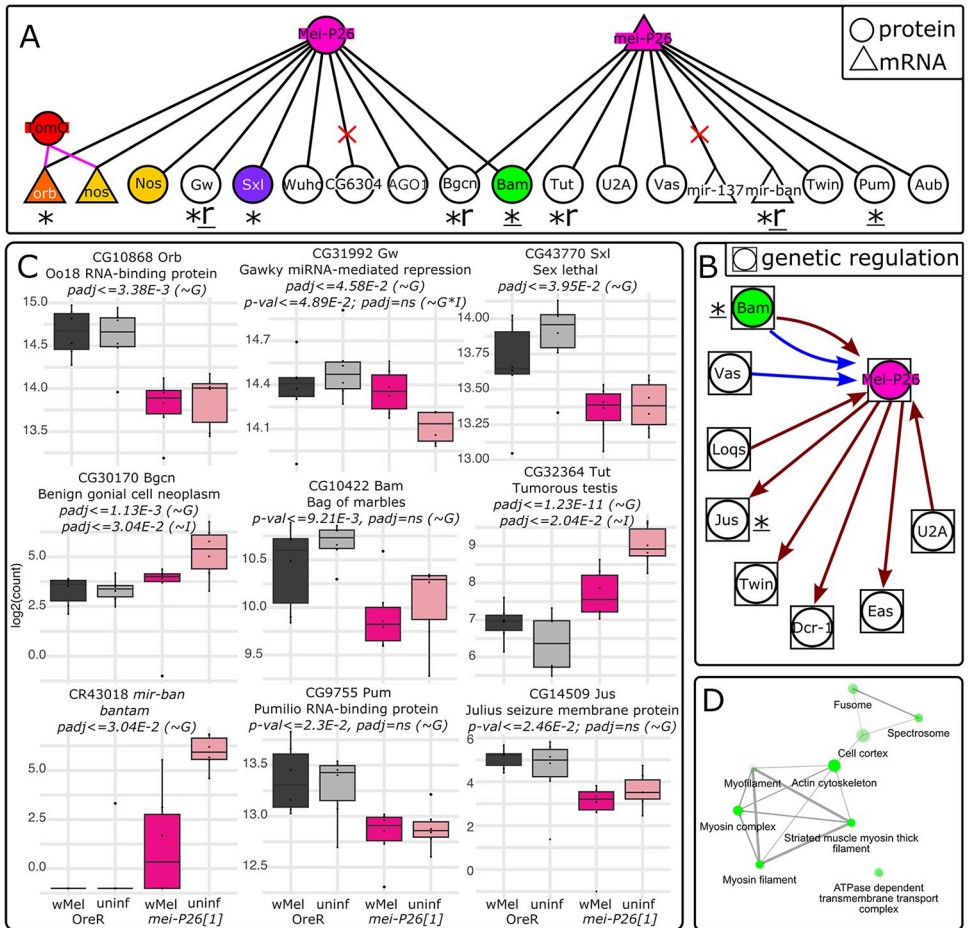

**Fig 8. Mei-P26 interactions predicted from the literature are supported by our ovary dual-RNAseq results.** (A, B) Predicted *mei-P26* (A) protein and mRNA physical interactions and (B) genetic interactions. Symbols near gene names indicate significant DE interactions: * = significance by genotype; r = significance by infection, indicating a rescue phenotype; _ = significance by uncorrected *p*-values only. Red "X"s in A indicate genes that were not expressed in our dataset (CG6304 and mir-137). (C) Normalized transcript count plots for the genes designated with symbols in A and B. In all barplots, dark gray = *w*Mel-infected OreR, light gray = uninfected OreR, dark pink = *w*Mel-infected *mei-P26 [1]*, light pink = uninfected *mei-P26[1]*. (D) Cellular component GO terms for the 87 *(~G*I)* DE genes reveal an enrichment for early germline cytoskeletal structures and membrane proteins. Wald test association for listed padj and *p*-values are given in parentheses, as in Fig 7: genotype (~G), infection (~I), genotype*infection (~G*I). The data underlying this figure can be found at NCBI, under BioProject number PRJNA1007602. OreR, Oregon R.

Binning gene expression patterns revealed that there are at least 6 main rescue gene expression phenotypes induced by *w*Mel-infection in mutant ovaries: full rescue, overshoot, undershoot, wild-type effect, inverse, and novel (Figs 7F–7K, 8A, 8C, S11A, S11B, S12A and S12B). In "full rescue" phenotypes (Figs 7F and S11A), *mei-P26[1]* *w*Mel gene expression is indistinguishable from OreR levels and significantly different from uninfected *mei-P26[1]* levels. "Overshoot" phenotypes (Fig 7G) exhibit expression that is significantly in the opposite direction of the uninfected *mei-P26[1]* knockdown effect relative to OreR expression. Oppositely, "under-shoot" phenotypes (Fig 7H) are partial corrections towards OreR expression in *mei-P26[1]* *w*Mel ovaries. The one "wild-type effect" gene, an ER and Golgi cytoskeletal membrane anchor protein, demonstrated differential expression only between *w*Mel-infected and uninfected OreR ovaries (Fig 7I). Five genes exhibit "inverse" differential expression phenotypes, such as a sodium symporter (Fig 7J) and the transcription factor Chronophage (S12A Fig), in which

*mei-P26[1]* and OreR ovaries show opposite changes in differential expression relative to the uninfected genotype. In total, only these 6 "wild-type effect" and "inverse" genes were differentially expressed due to infection in OreR ovaries. Lastly, "novel" genes (Fig 7K) are those that only exhibit differential expression in *w*Mel-infected *mei-P26[1]* ovaries, suggesting a novel rescue or compensation pathway. For example, *suppression of retinal degeneration disease 1 upon overexpression 2* (*sordd2*) ubiquitin ligase mRNA expression is up-regulated, without rescue, in both infected and uninfected *mei-P26[1]* ovaries (Fig 7E). One of the novel rescue candidates, SKP1-related C (*skpC*), is a down-regulated component of SCF ubiquitin ligase (Fig 7K), which might compensate for the up-regulation of *sordd2.*

GO terms associated with infection and the interaction between infection and genotype (Figs S11C–S11G, S12C–S12G and 8D) were consistent with *w*Mel's known ability to associate with host actin [67], microtubules, motor proteins [68–70], membrane-associated proteins [71], and chromatin [72]. Infection-associated genes were enriched in GO terms suggesting a cytoskeletal developmental function with chromatin interactions (S11C–S11G Fig). Across biological processes, cellular compartment, and molecular function categories, genes were enriched in terms suggesting interactions with actin, myosin, contractile fibers, and muscle cell differentiation. Chromatin interactions were indicated by associations with condensin complexes, centromeres, and mitotic chromosomes. Genotype-by-infection interaction-associated genes were also enriched in cytoskeletal and chromatin-interaction GO terms, in addition to plasma membrane/cell junction-associated terms such as cell junction and synapse structure maintenance, cell cortex, and lipid and calmodulin binding (S12C–S12G Fig).

Interestingly, 2 chorion proteins (Fig 7F), an ecdysteroid 22-kinase associated protein (S11A Fig), and an estrogen-related receptor (ERR) (S12A Fig), which are likely involved in chorion formation [73], were rescued by *w*Mel. Considering that we have found *mei-P26[1]* embryos to have weaker corions then OreR wild-type embryos when treated with bleach, this finding suggests a novel chorion-associated function for *mei-P26* and a novel rescue phenotype for *w*Mel.

None of the 663 transcribed *w*Mel genes were significantly differentially expressed between the OreR and *mei-P26[1]* host genotypes, after *p*-values were FDR-corrected (Figs 7D, S13 and S14 and S20 Table), suggesting host regulation by *w*Mel's constitutively expressed genes. On average, *w*Mel transcriptomes were sequenced at 7.8 to 42.4× depth of coverage and *D. melanogaster* transcriptomes were sequenced at 25.9 to 35.9× (S16 Table). While these coverage ranges overlap, the 663 *w*Mel genes detected only spanned −1.5-fold depletion to 1.3-fold up-regulation (S20 Table). In contrast, the 10,720 expressed *D. melanogaster* genes spanned an order of magnitude more differential expression, with −10.041-fold depletion to 9.866-fold up-regulation (S17–S19 Tables). Thus, *w*Mel genes are weakly differentially transcribed compared to *D. melanogaster* genes, suggesting that *w*Mel may alter host phenotypes through downstream impacts of constitutively expressed proteins. To explore the constitutive effects of *w*Mel expression, we tested for GO pathway enrichment in the expressed transcriptome (S13 Fig). Pathways for DNA replication, protein expression, membrane transport, and heterotrophy (e.g., oxidative phosphorylation) were detected in abundance. While many of these genes are likely involved in normal bacterial cell maintenance and replication, a significant enrichment in pathways for oxidative phosphorylation, translation, transcription, and membrane functions (S13A Fig) suggest that some of these genes could be co-opted for host manipulation or processing host resources.

Among the 1,044 significantly differentially regulated *D. melanogaster* genes impacted by *mei-P26* knockdown and infection were 6 genes (of 24) predicted in the literature (padj< = 0.05 Wald test and 9 of 24 by *p*-value< = 0.05; Figs 8 and S1 and S1 and S2 Tables). Loss of *mei-P26* function caused dysregulation in *oo18 RNA-binding protein* (*orb), gawky* (*gw), sex-*

*lethal (sxl)*, *benign gonial cell neoplasm (bgcn)*, *tumorous testis (tut)*, and *bantam (mir-ban)* (padj< = 0.05 Wald test, Fig 8C and * in Fig 8A and 8B). *Bam*, *Pumilio (pum)*, and *Justice Seizure (jus)* expression may be altered by *mei-P26* knockdown, but additional samples are needed to resolve high infection variance (e.g., *mir-ban*) and low differential expression change (e.g., *gw*) (*p*-value< = 0.05 Wald test, Fig 8C and "*" in Fig 8A and 8B). Two of 6 genes exhibited evidence of *w*Mel rescue with an "undershoot" rescue phenotype ("r" in Fig 8A and 8B): *bgcn* and *tut* by padjust (Fig 8C). If we consider genes with significant unadjusted *p*-values, *gw* may also exhibit evidence of "undershoot" rescue (Fig 8C, "r" in Fig 8A). Bantam or *mir-ban* expression suppression may be rescued as well, but high variance among *w*Mel *mei-P26[1]* ovaries precludes DE rescue detection with this dataset (Fig 8C). In contrast, *orb*, *sxl*, *bam*, and *nos* transcript levels are not altered by *mei-P26* knockdown or *w*Mel infection, and therefore must be regulated posttranscriptionally. In total, these differential expression results reinforce and expand our understanding of how *w*Mel rescues and reinforces host reproduction at transcriptional and posttranscriptional levels.

## Discussion

The *w*Mel strain of *Wolbachia* is a successful and widespread symbiont: naturally in *D. melanogaster* populations [19,20] and novelly through lab-generated trans-infections of *w*Mel into non-native hosts used for biological control [3]. However, we know remarkably little about the processes that shape how symbiont and host coevolve and find mutually beneficial mechanisms for their reproduction. In this work, we confirm that *w*Mel's CI strength is typically weak in *D. melanogaster*, (Fig 9A–9D, [14–18]) and show that *w*Mel confers benefits on host fertility through reinforcing host GSC maintenance and gamete differentiation (Figs 1–8).

Here, we demonstrate that *w*Mel infection can reinforce host fertility in both OreR wild-type and *mei-P26* mutants by correcting perturbed gene expression patterns, and we shed light on the long-standing mystery of how *w*Mel beneficially impacts host reproduction. Previously, *w*Mel was shown to rescue the partial loss of Sxl [31] and Bam [32], 2 essential genes for 2 very different stages of early oogenesis: GSC maintenance and CB differentiation and TA-mitosis, respectively. These findings left unresolved how *w*Mel suppresses these mechanistically and temporally distinct cellular processes and suggested that *mei-P26* was not involved [31]. We show with an array of alleles and developmental assays that infection with *w*Mel rescues defects associated with mutant *mei-P26*, a gene required for both GSC maintenance and differentiation. Infection with *w*Mel rescues all stages of oogenesis in *mei-P26* RNAi knockdowns, as well as hypomorphic and null alleles (Figs 1–5, 7, 8, S4–S7 and S10–S12 and S4–S15 and S18 and S19 Tables). Mei-P26 is a TRIM-NHL protein that regulates gene expression via mRNA translational inhibition through the Nos mRNA-binding complex, interactions with the RNA-induced silencing complex (RISC), and protein ubiquitination [38,74]. Importantly, Mei-P26 interacts with Sxl in the GSC and CB and Bam in the CB and cystocytes [36,50]. These interactions suggest that the mechanism *w*Mel employs to rescue *mei-P26* function may also be responsible for rescuing aspects of Sxl-dependent GSC maintenance and CB differentiation and Bam-dependent cyst differentiation.

In this in-depth molecular and cellular analysis of *w*Mel-mediated manipulation of GSC maintenance and differentiation, we discovered that *w*Mel rescues *mei-P26* germline defects through correcting perturbed pMad, Sxl, Bam, and Orb protein expression and *bgcn* and *tut* mRNA expression towards OreR wild-type levels (diagrammed in Fig 9A). Infection with *w*Mel partially rescued GSC-specific pMad expression relative to wild-type flies (Fig 2A–2E), suggesting that the bacteria restored BMP signaling from the somatic GSC niche [38]. Finding that these cells are also able to divide (Fig 2F–2J), further supports their functionality and

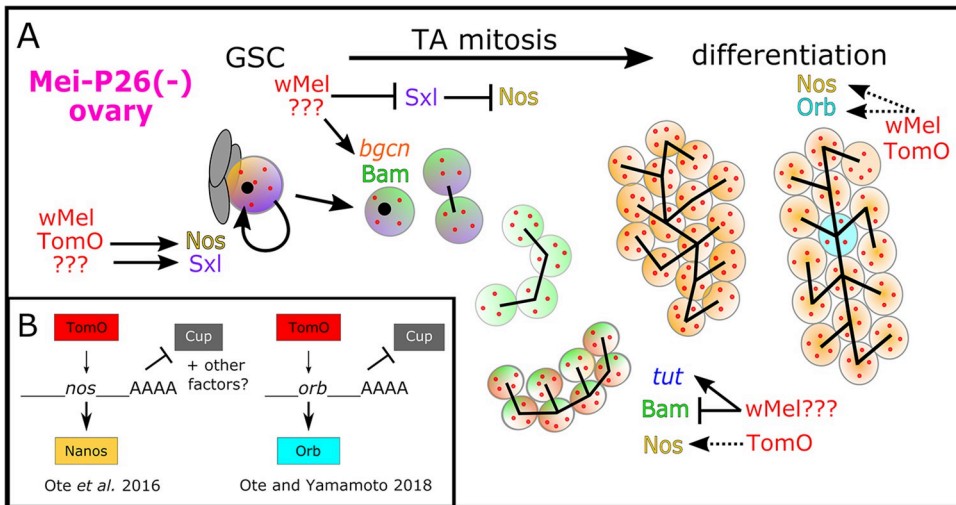

**Fig 9. Model of *w*Mel's interactions with essential host genes in early GSC maintenance and germline cyst formation.** (A) In *mei-P26* mutants, *w*Mel restores normal Bam, Sxl, and Orb expression through various mechanisms, most of which are unknown (??? = unknown *w*Mel factors). Protein and mRNA expression rescue correlates with cytological rescue from GSC maintenance through germline cyst differentiation. (B) Sxl rescue in the GSC is mediated through TomO's interaction with RNPs containing *nos* mRNA, up-regulating Nos translation. Later, in stage 8 of cyst oogenesis, TomO was found to bind *orb* mRNA, displacing the translational repressor Cup, and up-regulating Orb translation [34,35]. Dashed arrow in A indicates TomO's predicted function in enabling Orb derepression in early cyst differentiation. The factor that restores *mei-P26* repression of Orb translation in *mei-P26* mutants has not been identified. GSC, germline stem cell; RNP, ribonucleoprotein; TomO, toxic manipulator of oogenesis.

stemness. Given that *mei-P26* is likely involved in *Drosophila* Myc (dMyc) regulation, and overexpression of dMyc induces competitive GSCs [75], *w*Mel's GSC rescue mechanism may involve dMyc up-regulation. Although, our transcriptomic data indicate that dMyc is not significantly repressed at the transcriptional level (S15 Fig). In the germline cells that left the niche, *w*Mel recapitulated the properly timed changes in pMad, Sxl, and Bam expression that are normally influenced by *mei-P26* and are required during cystoblast differentiation to form the 16-cell germline cyst [39,40,50,74] (Figs 3, 4, and S7E–S7F). While Bam down-regulation is not fully rescued by wMel (Fig 4), finding that *tut* mRNA expression is partially reduced in *w*Mel-infected *mei-P26[1]* germaria (Fig 8) suggests that there may be an unreported regulatory interaction between Tut, Bam, and Mei-P26 in females (as in males [76]), which could rescue the overexpression of Bam in late germline cyst development.

This is the first time *w*Mel has been found to rescue oocyte differentiation post-cyst formation, suggesting that *w*Mel has pervasive impacts throughout oogenesis that may reinforce embryogenesis. We confirmed the downstream consequences of Bam expression rescue in *w*Mel-infected *mei-P26[1]* ovaries by finding fewer germline cyst tumors in infected flies (Fig 5A–5I). These germline cysts appeared to be functional, based on restored oocyte-specific expression of the essential oocyte differentiation protein Orb in infected *mei-P26[1]* cysts, relative to uninfected cysts (Fig 5J–5R). Finding that *w*Mel rescues Cp16 and Cp19 chorion protein, *tut*, *bgcn*, and other transcript expression in ovary tissues (Figs 7F, 8, S11 and S12) further supports the conclusion that *w*Mel can rescue the downstream impacts of *mei-P26* loss. Future work is needed to explore how the overshoot and novel DE categories mitigate the loss of *mei-P26*, which may occur through some alternate or compensatory pathway. In addition to the mechanism discussed previously for ubiquitin ligase rescue, the loss of Ionotropic receptor 76a (Ir76a) or nicotinic Acetylcholine Receptor α5 (nAChRα5) expression (S10A Fig) could be

rescued by the novel up-regulation of the Wunen-2 lipid phosphate phosphatase as a receptor or membrane transporter in wMel-infected *mei-P26[1]* ovaries (S12B Fig). Functions like these could underlie wMel-induced developmental resilience in OreR embryogenesis, as demonstrated by elevated hatch rates in wMel-infected OreR flies (Figs 1C and 6C).

The wMel strain's rescue of *D. melanogaster mei-P26* is likely independent of the *Wolbachia* TomO protein. Prior work on TomO indicates that the bacterial protein's primary mode of action is through destabilizing ribonucleoprotein (RNP) complexes [35] (diagrammed in Fig 9B). This is how TomO elevates Nos expression in the GSC [34] and how TomO inhibits the translational repressor Cup from repressing Orb translation in mid-oogenesis (stage 8) [35,77]. In contrast, Mei-P26 interacts with Ago1 RISC to inhibit Orb translation through binding the *orb' 3-UTR miRNA*-binding sites [38]. Although Mei-P26 and TomO produce opposite outcomes for Orb expression and TomO is not significantly up-regulated in *mei-P26 [1]* ovaries (S14C Fig), TomO-*orb* binding may underlie wMel's ability to rescue Orb repression, if wMel produces other factors that modulate the interaction. Alternatively, wMel may make another RNA-binding protein. Given that Sxl is also repressed through the *sxl* 3′-UTR by Bruno [78], such a protein could be involved in restoring Sxl regulation in *mei-P26* mutants. Neither *orb* nor *sxl* expression is rescued at the mRNA level by wMel infection (Fig 8C), lending further support for rescue at the translational or protein interaction/modification level.

Additional bacterial factors besides TomO must be involved in reinforcing host fecundity in wild-type flies and fertility mutants. First, Mei-P26 encodes 3 functional domains that confer at least 3 different functional mechanisms for modulating gene expression (mRNA-binding, miRNA-mediated, and ubiquitination). In a bacterial genome, these domains/functions are likely to be encoded by more than 1 protein [79]. Second, TomO rescues Sxl's GSC maintenance functions by binding *nos* mRNA, increasing levels of Nos translation, but it cannot fully rescue Sxl loss [34]. Immediately after the GSC divides to produce the CB, Sxl interacts with Bam, Mei-P26, Bgcn, and Wh to inhibit Nos translation [36,74]. Thus, a wMel factor that inhibits Nos translation, counteracting TomO's function to destabilize Nos translational repressors through RNA binding [34,35], remains to be identified. Furthermore, Nos up-regulation cannot explain Bam rescue because Nos and Bam exhibit reciprocal expression patterns during TA mitosis [36]. Third, exacerbation of the tumor phenotype induced by *nos*:Gal4-driven Mei-P26 overexpression by wMel infection (Fig 5I) suggests that wMel synthesizes a protein that directly mimics Mei-P26's functions in differentiation and can reach antimorphic levels. Misregulated expression of a *mei-P26*-like factor could explain why wMel-infected *mei-P26* RNAi females produce more eggs and offspring than OreR wild-type flies (Fig 1A and 1B). While TomO may recapitulate some of Mei-P26's functions, such as *orb*-binding, at least one other factor is needed to recapitulate proper Orb, Nos, and Bam regulation.

Understanding how *Wolbachia* interacts with host development at the molecular level is essential to deploying these bacteria in novel hosts and relying on vertical transmission to maintain them in host populations. CI has been a powerful tool for controlling host populations [80] and driving *Wolbachia* to high infection frequencies [3]. The continuous presence of bacterial reproductive manipulators in host GSCs opens the opportunity for compensatory developmental functions to evolve, similar to what we have shown for *mei-P26*. While it is not known if any of the naturally occurring *mei-P26* alleles [32,81] confer loss of function that is remedied by the presence of *Wolbachia*, natural variation in developmental genes theoretically could have provided the selection pressure for wMel to evolve its beneficial reproductive functions. Given that selection will favor the loss of CI if conflicting beneficial impacts on host fertility are realized [8], it is imperative that we understand the mechanisms underlying beneficial reproductive manipulations such as fertility reinforcement.

## Materials and methods

Russell and colleagues *Wolbachia* endosymbionts manipulate GSC self-renewal and differentiation to enhance host fertility

### Candidate gene selection

Leveraging what is known about wMel's abilities to rescue essential maintenance and differentiation genes [31,32] with empirical data on protein–protein and protein–mRNA interactions among these genes (esyN [82] networks in Figs S1A–S1D and 8; references in S1 Table) and data on interactions between host genes and wMel titers [37], we identified the essential germline translational regulator *meiotic P26* (*mei-P26*) as a potential target of bacterial influence over GSC maintenance and differentiation pathways (S1A and S1B Fig). Loss of *mei-P26* in uninfected flies produces mild to severe fertility defects through inhibiting GSC maintenance, meiosis, and the switch from germline cyst proliferation and differentiation [38–40].

### *Drosophila* stocks and genetic crosses

Flies were maintained on white food prepared according to the Bloomington Drosophila Stock Center (BDSC) Cornmeal Food recipe (aka "white food," see https://bdsc.indiana.edu/information/recipes/bloomfood.html). The wMel strain of *Wolbachia* was previously crossed into 2 *D. melanogaster* fly stocks, one carrying the markers and chromosomal balancers w[1]; Sp/Cyo; Sb/TM6B, Hu and the other carrying the germline double driver: P{GAL4-Nos.NGT} 40; P{GAL4::VP16-Nos.UTR}MVD1. These infected double balanced and ovary driver stocks were used to cross wMel into the marked and balanced FM7cB;;;sv[spa-pol], null/hypomorphic mutants, and UAS RNAi TRiP lines to ensure that all wMel tested were of an identical genetic background. The *D. melanogaster* strains obtained from the BDSC at the University of Indiana were: Oregon-R-C (#5), w1118; P{UASp-mei-P26.N}2.1 (#25771), y[1] w[*] P{w [+mC] = lacW}mei-P26-1 mei-P26[1]/C(1)DX, y[1] f[1]/Dp(1;Y)y[+]; sv[spa-pol] (#25716), y [1] w[1] mei-P26[mfs1]; Dp(1;4)A17/sv[spa-pol] (#25919), y[1] sc[*] v[1] sev[21]; P{y[+t7.7] v [+t1.8] = TRiP.GL01124}attP40 (#36855). We obtained the UASp-mRFP/CyO (1-7M) line from Manabu Ote at the Jikei University School of Medicine. *Drosophila simulans* w[–] stocks infected with the Riv84 strain of wRi and cured with tetracycline were sourced from Sullivan Lab stocks [83,84]. All fly stocks and crosses were maintained at room temperature or 25°C on white food (BDSC Cornmeal Food) because the sugar/protein composition of host food affects *Wolbachia* titer [84].

Wild-type fertility stocks: Paired wMel-infected (OreR_wMelDB) and uninfected Oregon R (OreR_uninf) stocks were made by crossing males of Oregon-R-C to virgin females from paired infected and uninfected balancer stocks of the genotype w[1];CyO/Sp;Hu/Sb. To ensure that no differences arose between these 2 *D. melanogaster* genotypes during the balancer cross or subsequently, we backcrossed males of the OreR_wMelDB stock to females of the OreR_uninf stock for 10 generations and repeated egg lay and hatch assays (F10_OreR_uninf). Paired infected and uninfected nosGal4/CyO and nosGal4/Sb flies were made by crossing to paired infected and uninfected balancer stocks, as described above.

The genomic and phenotypic consequences of the *mei-P26* alleles studied here and previously (e.g., [31]) have been characterized [40]. The hypomorphic *mei-P26[1]* allele was created by the insertion of a P{lacW} transposon in the first intron of *mei-P26*, which codes for the RING domain. The mostly null *mei-P26[fs1]* allele tested by Starr and Cline in 2002 was generated by a second P{lacW} insertion in *mei-P26[1]'s* first insertion. The fully (male and female) null *mei-P26[mfs1]* allele arose by deleting the P{lacW} insertion from *mei-P26[1]*, along with approximately 2.5 kb of flanking sequence. According to Page and colleagues (2000), these 3

**Table 1. Fecundity cross genotypes, sexes, ages, and control crosses.**

| Fecundity cross category | Female age | Male age | Parallel control crosses |
|---|---|---|---|
| wMel-infected mei-P26 rescue | 3–7 days old | 3–7 days | Uninfected and wild-type |
| wMel-infected OreR wild-type | 3–7 days old | 3–7 days | Uninfected OreR wild-type and other "wild-type" fertility stocks (e.g., nos: Gal4/CyO) |
| Aged wMel-infected mei-P26 rescue | 2–25 days old | 3–7 days | Uninfected and wild-type |
| Aged wMel-infected OreR wild-type | 2–45 days old | 3–7 days | Uninfected OreR wild-type |
| CI in wMel-*D. melanogaster* OreR | 3–7 days old | 0 or 3–7 days (from 3- to 7-day-old mothers) | Rescue, reciprocal, and uninfected |
| CI in wRi-*D. simulans w[–]* | 3–7 days old | 0 or 3–7 days (from 3- to 7-day-old mothers) | Rescue |

CI, cytoplasmic incompatibility; OreR, Oregon R.

alleles form a series, with increasing severity: *mei-P26[1]* < *mei-P26[fs1]* (and other fs alleles) < *mei-P26[mfs1]*.

## Fecundity crosses

**Overview.** SLR performed the 3,002 fecundity crosses continuously between October 2020 and January 2022, in batches based on their eclosion date (Table 1). Infected and uninfected OreR lines were maintained continuously to (1) control the age of grandmothers of CI males; (2) regularly have OreR virgins for *mei-P26* and CI fecundity assays; (3) collect and age WT females for the aged fertility assay; and (4) obtain large sample sizes from *mei-P26* fertility mutants. The full fecundity dataset is contained in S2 Table and plotted in S3 Fig.

**Cross conditions.** Food (vials of Bloomington's white food recipe), laying media (grape food), and incubation conditions were made in-house in large batches monthly. Grape spoons: Approximately 1.5 mL of grape agar media (1.2× Welch's Grape Juice Concentrate with 3% w/v agar and 0.05% w/v tegosept/methylparaben, first dissolved at 5% w/v in ethanol) was dispensed into small spoons, allowed to harden, and stored at 4°C until use. Immediately prior to use in a cross, we added ground yeast to the surface of each spoon.

**Preparation of flies for fecundity crosses.** Paired wMel-infected and uninfected genetic crosses were performed in parallel for the 4 *mei-P26* mutants (RNAi, [mfs1], [1], [mfs1/1]) to produce homozygous virgin females of both infection states for parallel fecundity crosses. Mutant genetic crosses were performed multiple (3 or more) times across the 14 months to produce homozygotes from many different parents and matings. CI crosses were performed with males from infected grandmothers that were 3 to 7 days old. Males were aged either 0 (collected that day) or 5 days, depending on the cross. We collected males from vials over multiple days, so the majority of males were not the first-emerged.

Virgin female flies collected for fecundity crosses were transferred to fresh food and aged an average of 5 days at room temperature (from 3 to 7), or longer for the aged OreR, *mei-P26 RNAi*, and *mei-P26[1]* fecundity crosses. Males and virgin female flies were stored separately, and males were aged 0 or 3 to 7 days. Long-term aged virgin females were kept as small groups in vials of fresh white food. Every few days, the vials were inspected for mold and flies were moved on to fresh food.

**Fecundity cross protocol.** In the afternoon of the first day of each cross, 1 male and 1 female fly were knocked out on a $CO_2$ pad, added to a vial containing a grape spoon, allowed to recover, and then transferred to a 25°C constant humidity incubator on a 12 light/dark

cycle to allow for courting and mating. The following day, each spoon was replaced with a fresh spoon, and the vials were returned to 25˚C for more mating. On the third day, we removed the flies from the vials, counted the number of eggs laid on the spoon's grape media, and replaced the spoon in the vial at 25˚C for 2 days. After approximately 40 h, we counted the number of hatched and unhatched eggs. The exact times and dates for all steps in all crosses were recorded (S2 Table) and plotted to show consistent results across the time frame (S3 Fig).

**Fecundity cross analysis.** Fecundity data were parsed using perl scripts and plotted in R. Individual crosses were treated as discrete samples. Hatch rates were calculated from samples that laid 20 or more eggs (for 3- to 7-day-old female plots) and 10 or more eggs (for age versus hatch rate plots). Egg lay and offspring production rates were calculated from raw counts, divided by the fraction of days spent laying (metadata in S2 Table). Crosses with 0 laid eggs (and offspring) were not rejected, as the fertility mutants often laid 0 to few eggs. Our cross conditions were highly consistent and run daily, lending confidence to these zero-lay/offspring samples (S3 Fig). *Drosophila* fecundity versus female age data were fit in R using the stat_smooth loess method, formula y~x, with 0.95 level confidence interval.

## Non-disjunction assays

We placed females homozygous for the *mei-P26[1]* allele bearing the yellow mutation (y[1] w [*] mei-P26[1];;; sv[spa-pol]) in vials with males bearing a wild-type yellow allele fused to the Y chromosome (y[1] w[1] / Dp(1;Y)y+). We made 7 or 8 vials of 1 to 10 females mated to a count-matched 1 to 10 males for infected and uninfected *mei-P26* hypomorphs, respectively. After eclosion, we screened for non-disjunction in the progeny by the presence of yellow males (XO) and normally colored females (XXY). Rates of non-disjunction (NDJ) were calculated to account for the inviable progeny (XXX and YO) with the equation NDJ rate = 2 *NDJ offspring/all offspring = (2XXY + 2X0)/(XX+XY+2XXY+2X0), as in [41].

## Ovary fixation and immunocytochemistry

Within a day or 2 or eclosion, flies were transferred to fresh food and aged 3 to 7 days, or longer as indicated in the text. For long aging experiments, flies were transferred to fresh food every week and investigated for mold every few days. We dissected the ovaries from approximately 10 flies from each cross in 1× PBS and separated the ovarioles with pins. Ovaries were fixed in 600 μl heptane mixed with 200 μl devitellinizing solution (50% v/v paraformaldehyde and 0.5% v/v NP40 in 1× PBS), mixed with strong agitation, and rotated at room temperature for 20 min. Oocytes were then washed 5× in PBS-T (1% Triton X-100 in 1× PBS) and treated with RNAse A (10 mg/ml) overnight at room temperature.

After washing 6 times in PBS-T, we blocked the oocytes in 1% bovine serum albumin in PBS-T for 1 h at room temperature, and then incubated the oocytes in the primary antibodies diluted in PBS-T overnight at 4˚C (antibodies and dilutions listed below). The following day, we washed the oocytes 6 times in PBS-T and incubated them in secondary antibodies diluted 1:500 in PBS-T overnight at 4˚C. On the final day, we washed the oocytes a final 6 times in PBS-T and incubated them over 2 nights in PI mounting media (20 μg/ml PI (Invitrogen #P1304MP) in 70% glycerol and 1× PBS) at 4˚C. Overlying PI medium was replaced with clear medium, and oocytes were mounted on glass slides. Slides were stored immediately at −20˚C and imaged within a month. Infected and uninfected, as well as experimental and control samples were processed in parallel to minimize batch effects. Paired wMel-infected and uninfected OreR stocks were used as wild-type controls.

Primary monoclonal antibodies from the Developmental Studies Hybridoma Bank (University of Iowa) were used at the following dilutions in PBS-T: anti-Hts 1:20 (1B1) [45], anti-

Vas 1:50 [44], anti-Orb 1:20 (4H8) [63], anti-Bam 1:5 [36], and anti-Sxl 1:10 (M18) [85]. Primary monoclonal antibodies from Cell Signaling Technology were used at the following dilutions: anti-Phospho-Smad1/5 (Ser463/465) (41D10) 1:300 (#9516S) and anti-Phospho-Histone H3 (Ser10) Antibody 1:200 (#9701S). Anti-*Wolbachia* FtsZ primary polyclonal antibodies were used at a 1:500 dilution (provided by Irene Newton). Secondary antibodies were obtained from Invitrogen: Alexa Fluor 405 Goat anti-Mouse (#A31553), Alexa Fluor 488 Goat anti-Rabbit (#A12379), and Alexa Fluor 647 Goat anti-Rat (#A21247).

## Confocal imaging

Oocytes were imaged on a Leica SP5 confocal microscope with a 63× objective. Optical sections were taken at the Nyquist value for the objective, every 0.38 μm, at variable magnifications, depending on the sample. Most germaria were imaged at 4× magnification and ovarioles and cysts were imaged at a 1.5× magnification. Approximately 10 μm (27 slices) were sampled from each germarium and 25 μm (65 slices) were sampled from each cyst for presentation and analysis.

PI was excited with the 514 and 543 nm lasers, and emission from 550 to 680 nm was collected. Alexa 405 was imaged with the 405 nm laser, and emission from 415 to 450 nm was collected. Alexa 488 was imaged with the 488 laser, and emission from 500 to 526 nm was collected. Alexa 647 was imaged with the 633 laser, and emission from 675 to 750 nm was collected.

## Image analysis

Germaria and ovarioles were 3D reconstructed from mean projections of approximately 10 μm-thick nyquist-sampled confocal images (0.38 μm apart) in Fiji/ImageJ. Germline cyst developmental staging followed the standard conventions established by Spradling [86] and the criteria described below. We also scored and processed the confocal z-stacks for analysis as follows in the sections below.

**GSC quantification.** GSCs were scored by the presence of pMad and an Hts-labeled spectrosome in cells with high cytoplasmic volumes adjacent to the somatic germ cell niche and terminal filament [56]. When antibody compatibility prevented both pMad and Hts staining (e.g., with anti-pHH3 staining), only one was used for GSC identification. The spectrosome was distinguished from the fusome by its position anterior of the putative GSC nucleus and the presence of a posterior fusome that continues into a posteriorly dividing cystoblast. We calculated the number of GSCs per germarium as the average number of pMad and Hts-spectrosome-expressing cells. Non-whole increments of GSCs indicate situations where a putative GSC expresses one attribute, but not the other.

**Mitotic GSC quantification.** Putative GSCs were identified as described above and scored for the presence of anti-pHH3 staining. The number of pHH3-positive cystocytes in region 1 was also quantified across germaria.

**Oocyte cyst tumor quantification.** Using a 10× objective, we manually counted the number of nurse cells in each stage 6 to 10b cyst. Each count was repeated 3 times consistently and the cyst tallied as having less than 15, 15, or more than 15 nurse cells per cyst.

**Bam and Sxl expression measured by fluorescence.** We summed 27-slice nyquist-sampled z-stacks of each germarium in Fiji/ImageJ. Fluorescence intensity was measured by setting ImageJ to measure: AREA, INTEGRATED DENSITY, and MEAN GRAY VALUE. Germarium regions were delimited as in [42] using Vas expression to indicate the germline. Three representative background selections were measured for subtraction. The corrected total cell fluorescence (CTCF) was calculated for each region of the oocyte as follows:

CTCF = Integrated Density–Area of selected cell * Mean fluorescence of background readings. We controlled for staining intensity within a germarium and compared relative values among germaria.

**Orb expression.** Anti-Orb-stained oocyte cysts were manually scored for stage and Orb oocyte-staining in confocal z-stacks ImageJ.

## Transcriptomics

We collected OreR and hypomorphic *mei-P26[1]* flies uninfected and infected with wMel and wMel-infected *nos:Gal4>UAS:mei-P26 RNAi* and nos:Gal4/CyO flies for RNAseq. Flies were collected after eclosion and moved to fresh food until they were 3 to 5 days old. Ovary dissections were performed in 1× PBS, in groups of 20 to 30 flies to obtain at least 10 mg of ovary tissue for each sample. After careful removal of all non-ovary somatic tissue, ovaries were promptly moved to RNAlater at room temperature, and then transferred to −80°C within 30 min for storage. Frozen tissue was shipped on dry ice to Genewiz Azenta Life Sciences for RNA extraction, cDNA synthesis, Illumina library preparation, and Illumina sequencing. The RNAi and control samples—both *nos:Gal4>UAS:mei-P26 RNAi* and *nos:Gal4/CyO* genotypes —were processed to cDNA with T-tailed primers, recovering only host transcripts. The *mei-P26[1]* and OreR samples were processed to cDNA using random hexamers and ribosomal sequences were depleted with sequential eukaryotic and bacterial rRNA depletion kits (Qiagen FastSelect). Illumina dual-indexed libraries were made from these cDNAs and sequenced as 2 × 150 bp reads.

We processed and analyzed RNAseq datasets for differential expression using standard computational approaches and custom perl parsing scripts. Briefly, following demultiplexing, we trimmed adapter fragments from the RNAseq reads with Trimmomatic [87]. To generate sitewise coverage data to examine read alignments directly, we used the STAR aligner [88] and samtools [89], and plotted read depths across samples in R. We quantified transcripts by pseudoalignment with Kallisto [90]. The choice of reference transcriptome was key to recovering the maximum number of alignments: we merged the NCBI RefSeq assemblies for the wMel reference genome CDSs and RNAs from genomic (accession GCF_000008025.1) and the *D. melanogaster* reference genome RNAs from genomic (accession GCF_000001215.4; Release_6_plus_ISO1_MT) to obtain alignments against the full, non-redundant host-symbiont transcriptome. Simultaneous mapping to both genomes was performed to avoid cross-species mismapping [91]. This reference transcriptome was indexed at a kmer length of 31 in Kallisto (version 0.45.1) [90] and reads were pseudoaligned against this reference with the "kallisto quant" command and default parameters. Host and symbionts have distinct transcriptome distributions [92], necessitating the separation of the 2 transcriptomes prior to transcript normalization and quantification in DESeq2 [93], which we performed with a custom perl script.

We imported the subset *D. melanogaster* and *w*Mel Kallisto transcriptome quantifications separately into R with Tximport [94] for DESeq2 analysis [93]. Transcript-level abundances were mapped to gene IDs to estimate gene-level normalized counts. For each transcriptome, we filtered out low-count/coverage genes across samples by requiring at least 5 samples to have a read count of 10 or more. We modeled interactions among our experimental groups as a function of genotype, infection, and the interaction between genotype and infection (~genotype + infection + genotype*infection) and performed Wald Tests to detect differential expression. Briefly, the maximum likelihood gene model coefficient for each gene's expression count was calculated and divided by its standard error to generate a z-statistic for each gene under the full and reduced models. These z-statistics were compared to the values obtained under standard normal distribution for *p*-value calculations. FDR/Benjamini–Hochberg corrections

were performed on these *p*-values to reduce the number of false positives. Normalized counts for each gene were output with the plotCounts() function in DESeq2 for plotting in R.

We manually investigated the top hits for each test for evidence of germline expression in the Fly Cell Atlas project [95] through Flybase. GO categories for differentially expressed genes returned from each DESeq2 Wald Test and tested for significance against the reference transcriptome set of IDs with ShinyGO0.77 [96].

Transcriptomic data generated in this study are available through NCBI BioProject number PRJNA1007602

## Plotting and statistical analysis

Fecundity data were plotted, analyzed, and statistics were calculated in R. Differences in lay, hatch, and offspring production rates were evaluated with the nonparametric Wilcoxon rank sum test. When this was infeasible because non-zero samples were so few (e.g., null alleles), we compared the binomially distributed categorical groups of females who laid or did-not-lay eggs with the Fisher's Exact test. All sample sizes, means, and *p*-values are presented in S2 Table. Fecundity was plotted across fly ages and fitted with local polynomial regression. Single age pots were made with base R and the beeswarm [97] package and fecundity-vs-time plots were made with the ggplot2 [98] package.

Confocal micrograph fluorescence intensities were analyzed and plotted, and statistics were calculated in R. Relative fluorescence intensities between oocyte genotypes were compared with the nonparametric Wilcoxon rank sum test. Counts of GSCs, mitotic GSCs, tumorous germline cysts, and orb-specific cysts were compared with Fisher Exact tests. Plots were made with the ggplot2 [98] package.

We analyzed and plotted the dual-RNAseq results from Kallisto and DESeq2 in R. Bar and scatter plots were made with the ggplot2 package [98] and volcano plots were made with the EnhancedVolcano package (release 3.17) [99].

## Supporting information

**S1 Fig. Key genetic regulators of germline stem cell (GSC) maintenance and differentiation in *D. melanogaster* oogenesis.** (A) Diagram of the GSC niche illustrating a subset of the genes that shift in expression in the cystoblast following GSC mitosis. (B) Model of the relative levels of protein expression during germline cyst development. (C, D) esyN interaction networks for (C) *sxl* and (D) *bam* gene products (supporting references in S1 Table).
(TIF)

**S2 Fig. *Drosophila mei-P26* genetic resources and gene expression characterization.** (A) Genomic map and gene model for *mei-P26* and the studied alleles. The insertion of a P{lacW} transposon in the first intron of *mei-P26[1]* impacts the RING domain. The *mei-P26[mfs1]* allele was generated by deletion of this insertion and 0.7–1.6 kb of DNA flanking each side of the insertion site. (B, C) mei-P26 transcript coverage and (D, E) Kallisto Kallisto normalized transcript counts for *D. melanogaster mei-P26* transcripts from (B, D) *mei-P26[1]* and OreR *w*Mel-infected vs. uninfected ovaries and (C, E) nos:Gal4>UAS:meiP26RNAi vs. OreR *w*Mel-infected ovaries. The data underlying this figure can be found at NCBI, under BioProject number PRJNA1007602.
(TIF)

**S3 Fig. Fecundity data acquisition plots vs. time.** (A, B) Female *mei-P26* RNAi, (C, D) Female *mei-P26[1]*, and (E, F) CI assays. Both (A, C, E) egg lay rates and (B, D, F) hatch rates were consistent over time, across genetic crosses, and across fecundity crosses. A factor of 0.1

was added to the y-axis values as an offset to see zero-lay and zero-percent hatch data points.
(TIF)

**S4 Fig. Infection with *w*Mel rescues *mei-P26* function in females and males.** (A, B) Hypo-morphic *mei-P26[1]* and (C, D) *nos:Gal4>mei-P26RNAi D. melanogaster* female fecundity vs. age, fit with a local polynomial regression (dark gray bounds = 95% confidence intervals). Infection with *w*Mel elevates offspring production across the female lifespan through increasing the number of eggs laid and the proportion of those eggs that hatch. (E–G) Male *mei-P26* rescue: *w*Mel infection produced significantly higher rates of (D) overall offspring production, broken into (F) egg lay and (F) egg hatch, in RNAi, hypomorphic, and null *mei-P26[1]* knock-down male flies mated to wild-type females of the same age and infection status. Wilcoxon rank sum * = $p < 0.05$, ** = 0.01, **** = 1e-4. The data underlying this figure can be found on Dryad at doi.org/10.7291/D1DT2C. (H, I) Homozygous hypomorphic *mei-P26[1]* stocks (H) infected with *w*Mel *Wolbachia* or (I) uninfected. Mold growth (green food vs. tan/brown food) is uninhibited in the uninfected stocks due to embryo and larval death, which both feeds and fails to stop mold. Infection enables stable robust stock persistence because larval production outruns mold growth. Both stocks were started at the same time (see 12/28 on the label). The *w*Mel-infected stock never needed any adults added, whereas the uninfected stock produced too few offspring and had to be supplemented at every vial flip to keep the stock going artificially. We ended this after a few months and the uninfected stock fully died out.
(TIFF)

**S5 Fig. Hypomorphic *mei-P26[1]* ovarioles and germaria exhibit a range of (A–I) wMel-infected and (J–R) uninfected phenotypes.** Red = PI DNA staining, yellow = anti-Vas staining, and cyan = anti-Hts staining. Scale bars A, D, G, H, I, J–L, N, O, Q, R = 50 μm; B, C, E, F, M, P = 25 μm.
(TIF)

**S6 Fig. Infection with *w*Mel does not rescue *mei-P26's* function in meiosis.** (A) Table containing X-chromosome nondisjunction experimental data. (B) Beeswarm boxplot of the rate of X-chromosome non-disjunction (NDJ) in each experiment. There was no significant difference between infected and uninfected *mei-P26[1]* females.
(TIF)

**S7 Fig. GSC maintenance and germline differentiation are rescued by *w*Mel infection.** (A, B) Confocal mean projections of *D. melanogaster* germaria stained with antibodies against Hts and pMad. RNAi knockdown of *mei-P26* does not affect GSC maintenance (Fig 2E). (C) Violin plots of the number of GSCs per germarium in 10- to 13-day-old females. As fully functional GSCs express pMad and have Hts-labeled spectrosomes, each was weighted by half and allows for partial scores. Wilcoxon rank sum * = $p < 0.05$, ** = 0.01. (D) F-K) Violin plots of the number of mitotic cystocytes per germarium detected by pH3 expression. (E, F) Bar-scatter plots of total (E) Sxl and (F) Bam fluorescence expression levels across the germarium, by region. (G) 1D barplots of oocyte-specific Orb staining among germline cysts, distributed across cyst developmental stages. The data underlying this figure can be found on Dryad at doi.org/10.7291/D1DT2C.
(TIF)

**S8 Fig. The *w*Mel strain of *Wolbachia* is a beneficial manipulator of host reproduction.** (A–C) Beeswarm boxplots showing that *w*Mel infection elevates wild-type *D. melanogaster* fertility relative to uninfected flies of the same genotype. (A) Overall offspring production, (B) egg lay, and (C) egg hatch were variably impacted in different "wild-type" genotypes. (D) *D.*

*melanogaster* eggs laid per female per day plot against female age, fit with a local polynomial regression (dark gray bounds = 95% confidence intervals). The data underlying this figure can be found on Dryad at doi.org/10.7291/D1DT2C.
(TIF)

**S9 Fig. Cytoplasmic incompatibility (CI) differs in strength between *Drosophila-Wolbachia* associations and weakens with male age.** (A–D) Beeswarm box plots of (A, C) egg hatch rate and (B, D) offspring production of uninfected and *w*Mel-infected *D. melanogaster* OreR females mated to (A, B) zero-day-old and (C, D) 5-day-old *w*Mel-infected males. (E–H) Beeswarm box plots of (E, G) egg hatch rate and (F, G) offspring production in uninfected and wRi-infected *D. simulans* females mated to (A, B) zero-day-old and (C, D) 5-day-old wRi-infected males. Wilcoxon rank sum * = $p < 0.05$, ** = 0.01, *** = 0.001, **** = 1e-4. The data underlying this figure can be found on Dryad at doi.org/10.7291/D1DT2C.
(TIF)

**S10 Fig. *D. melanogaster* genes significantly differentially expressed due to genotype (Wald Test ~*G* vs. ~*G*+*I*+*G*\**I*) reveal that *mei-P26* is required for the regulation of many genes.** (A) Kallisto normalized transcript counts for *D. melanogaster* genes (top 15 hits (including Fig 7E) *padj< = 2.0E-30*; see Figs 7 and S2 for other plots). Barplots are colored by group: dark gray = *w*Mel-infected OreR, light gray = uninfected OreR, dark pink = *w*Mel-infected *mei-P26[1]*, light pink = uninfected *mei-P26[1]*. (B–G) GO analysis for mei-P26-associated DE genes reveal an enrichment for processes involving chromatin, recombination, protein–protein interactions, and muscle cell differentiation. GO enrichment (B–D) category plots and (E–G) term interaction networks for the categories of (B, E) biological process, (C, F) cellular component, and (D, G) molecular function. The data underlying this figure can be found at NCBI, under BioProject number PRJNA1007602.
(TIF)

**S11 Fig. *D. melanogaster* genes significantly differentially expressed due to infection state (Wald Test ~*I* vs. ~*G*+*I*+*G*\**I*) reveal an enrichment of rescue events, genes for which *w*Mel-infected *mei-P26[1]* ovaries exhibit OreR expression levels.** (A, B) Kallisto normalized transcript counts for *D. melanogaster* genes exhibiting (A) rescue and (B) overshoot of OreR expression levels (padj< = 0.002; see Fig 7 for other plots). Barplots are colored by group: dark gray = *w*Mel-infected OreR, light gray = uninfected OreR, dark pink = *w*Mel-infected *mei-P26[1]*, light pink = uninfected *mei-P26[1]*. (C–G) GO analysis for infection DE genes reveal an abundance of cytoskeletal and chromatin components. GO enrichment (C–E) category plots and (F, G) term interaction networks for the categories of (C, F) biological process, (D, G) cellular component, and (E) molecular function (no network for a single term). The data underlying this figure can be found at NCBI, under BioProject number PRJNA1007602.
(TIF)

**S12 Fig. *D. melanogaster* genes significantly differentially expressed due to the joint infection-by-genotype state (Wald Test ~G\*I vs. ~G+I+G\*I) reveal an enrichment of reversal DEG, genes for which *w*Mel-infected ovaries exhibit inverse DE patterns for *mei-P26[1]* and OreR ovaries.** (A–C) Kallisto normalized transcript counts for *D. melanogaster* genes exhibiting (A) inverse, (B) undershoot, and (C) novel regulation of OreR expression levels (padj< = 0.02; see Fig 7 for other plots). Barplots are colored by group: dark gray = *w*Mel-infected OreR, light gray = uninfected OreR, dark pink = *w*Mel-infected *mei-P26[1]*, light pink = uninfected *mei-P26[1]*. (D–G) GO analysis for infection DE genes reveal cytoskeletal and membrane factors. GO enrichment (C–E) category plots and (F, G) term interaction networks for the categories of (C, G) biological process, (D) cellular component (see Fig 8D for

network), and (E, F) molecular function. The data underlying this figure can be found at NCBI, under BioProject number PRJNA1007602.
(TIF)

**S13 Fig. wMel *Wolbachia* transcriptome differential expression GO categories of all expressed genes reveal cell maintenance, central metabolic, and membrane-associated processes.** GO enrichment (A) category plot, (B) hierarchical clustering tree, and (C) term interaction network the biological process category. The data underlying this figure can be found at NCBI, under BioProject number PRJNA1007602.
(TIFF)

**S14 Fig. Kallisto normalized transcript counts for *w*Mel *Wolbachia* (A, B) differential expression candidate genes (pending deeper sampling) and (C) genes of interest from the literature.** Barplots are colored by group: dark gray = *w*Mel-infected OreR, light gray = uninfected OreR, dark pink = *w*Mel-infected *mei-P26[1]*, light pink = uninfected *mei-P26[1]*. Wald test genotype-association *p*-values and adjusted *p*-values (padj). The data underlying this figure can be found at NCBI, under BioProject number PRJNA1007602.
(TIF)

**S15 Fig. Nonsignificant Kallisto normalized transcript counts for a subset of essential *D. melanogaster* oogenesis genes from the literature selected based upon their known interactions with *mei-P26*, *sxl*, *bam*, or germ plasm formation (a mid-stage 9 of oogenesis).** Barplots are colored by group: dark gray = *w*Mel-infected OreR, light gray = uninfected OreR, dark pink = *w*Mel-infected *mei-P26[1]*, light pink = uninfected *mei-P26[1]*. The data underlying this figure can be found at NCBI, under BioProject number PRJNA1007602.
(TIF)

**S1 Table. esyN references for *sxl* and *bam* interactions in S1C Fig and *mei-P26* interactions in Fig 9.**
(PDF)

**S2 Table. Annotated references used to make S1A and S1B Fig and the table in Fig 9C.**
(PDF)

**S3 Table. Full fecundity dataset (*n* = 3,002).** See S3 Table.
(TSV)

**S4 Table. Fecundity statistics: offspring produced per female per day in single female-by-single male crosses.** Experimental genotypes, infection statuses, and sexes are listed. The mate for each cross was OreR, of the same infection status, and of the opposite sex as the experimental fly. Males were aged 3–6 days, except for the young male CI crosses, which were aged zero days (distinguished with "-0d" and "-5d" labels). *P*-values <0.01 are in light green and <0.05 are in dark green for clarity.
(PDF)

**S5 Table. Fecundity statistics: eggs produced per female per day in single female-by-single male crosses.** Experimental genotypes, infection statuses, and sexes are listed. The mate for each cross was OreR, of the same infection status, and of the opposite sex as the experimental fly. Males were aged 3–6 days, except for the young male CI crosses, which were aged zero days (distinguished with "-0d" and "-5d" labels). *P*-values <0.01 are in light green and <0.05 are in dark green for clarity.
(PDF)

**S6 Table. Fecundity statistics: percentage of eggs that hatched from single female-by-single male crosses that laid > = 20 eggs.** Experimental genotypes, infection statuses, and sexes are listed. The mate for each cross was OreR, of the same infection status, and of the opposite sex as the experimental fly. Males were aged 3–6 days, except for the young male CI crosses, which were aged zero days (distinguished with "-0d" and "-5d" labels). Sample counts (n1, n2) in parentheses are for Fisher exact tests (samples with hatched eggs vs. no hatched eggs, opposed to % hatch for samples with 20 or more eggs laid). *P*-values <0.01 are in light green and <0.05 are in dark green for clarity.
(PDF)

**S7 Table. Fecundity versus age statistics.**
(PDF)

**S8 Table. Germline stem cell (GSC) counts per germarium.**
(PDF)

**S9 Table. Number of GSCs in mitosis (anti-pHH3-positive staining), per germarium.**
(PDF)

**S10 Table. Number of cystocytes (CC) in mitosis (anti-pHH3-positive staining), per germarium.**
(PDF)

**S11 Table. Sxl expression by germarium region, measured by fluorescence intensity.**
(PDF)

**S12 Table. Bam expression by germarium region, measured by fluorescence intensity.**
(PDF)

**S13 Table. Relative Bam vs. pMad expression, measured by fluorescence, in GSCs.**
(PDF)

**S14 Table. Tumorous germline cyst counts.** Normal cysts contain 16 germline-derived cells, 15 nurse cells and 1 oocyte. Cysts containing greater or less than 15 nurse cells were scored as tumorous or abnormal, respectively.
(PDF)

**S15 Table. Counts of germline cysts exhibiting oocyte-specific Orb expression (Y), unclear staining (M), or no specific expression, indicating developmentally abnormal cysts lacking specified oocytes.**
(PDF)

**S16 Table. Transcriptomic dataset generated to test the impacts of *mei-P26* knockdown and *w*Mel infection.** Data deposited under NCBI BioProject number PRJNA1007602.
(PDF)

**S17 Table. *D. melanogaster* genes Wald Test significant results for ~Genotype vs. ~Genotype+Infection+Genotype\*Infection.**
(PDF)

**S18 Table. *D. melanogaster* genes Wald Test significant results for ~Infection vs. ~Genotype+Infection+Genotype\*Infection.**
(PDF)

**S19 Table.** ***D. melanogaster* genes with Wald Test significant results for ~Genotype\*Infection vs. ~Genotype+Infection+Genotype\*Infection.**
(PDF)

**S20 Table.** *w*Mel Wolbachia genes Wald Test significant results for ~Genotype vs. ~1.
(PDF)

## Acknowledgments

We thank the UCSC Life Sciences Microscopy Center (RRID:SCR_021135) and Ben Abrams for training and the use of their microscopes. We thank Irene Newton for aliquots of anti-*Wolbachia* FtsZ antibodies and Manabu Ote for a stock of UASp-mRFP/CyO (1-7M) flies. We thank Russ Corbett-Detig for his comments and feedback throughout this project.

## Author Contributions

**Conceptualization:** Shelbi L. Russell, William T. Sullivan.

**Data curation:** Shelbi L. Russell, Jennie Ruelas Castillo.

**Formal analysis:** Shelbi L. Russell, Jennie Ruelas Castillo.

**Funding acquisition:** Shelbi L. Russell, William T. Sullivan.

**Investigation:** Shelbi L. Russell.

**Methodology:** Shelbi L. Russell.

**Project administration:** Shelbi L. Russell.

**Resources:** Shelbi L. Russell.

**Software:** Shelbi L. Russell.

**Supervision:** Shelbi L. Russell, William T. Sullivan.

**Validation:** Shelbi L. Russell.

**Visualization:** Shelbi L. Russell.

**Writing – original draft:** Shelbi L. Russell.

**Writing – review & editing:** Shelbi L. Russell, Jennie Ruelas Castillo, William T. Sullivan.

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
