## [Editor Report · Decision Letter 0]

26 Jan 2023

Dear Dr Russell, 

Thank you for submitting your manuscript entitled "Wolbachia endosymbionts manipulate GSC self-renewal and differentiation to enhance host fertility" for consideration as a Research Article by PLOS Biology.

Your manuscript has now been evaluated by the PLOS Biology editorial staff as well as by an academic editor with relevant expertise and I am writing to let you know that we would like to send your submission out for external peer review. Although we find the phenomenon reported here interesting, I should note that we have yet to make a firm call about whether the study provides a sufficient advance for PLOS Biology, as there is some question about the extent that this represents the basis for emerging mutualism and as the mechanism of wMel's effects is not fully elucidated. We will be looking for enthusiasm from the reviewers regarding the fit of your study for PLOS Biology. 

Before we can send your manuscript to reviewers, we need you to complete your submission by providing the metadata that is required for full assessment. To this end, please login to Editorial Manager where you will find the paper in the 'Submissions Needing Revisions' folder on your homepage. Please click 'Revise Submission' from the Action Links and complete all additional questions in the submission questionnaire.

Once your full submission is complete, your paper will undergo a series of checks in preparation for peer review. After your manuscript has passed the checks it will be sent out for review. To provide the metadata for your submission, please Login to Editorial Manager (https://www.editorialmanager.com/pbiology) within two working days, i.e. by Jan 30 2023 11:59PM.

Kind regards,

Lucas

Lucas Smith, Ph.D.

Associate Editor

PLOS Biology

lsmith@plos.org

---

## [Decision Letter · Decision Letter 1]

27 Mar 2023

Dear Dr Russell,

Thank you for your patience while your manuscript "Wolbachia endosymbionts manipulate GSC self-renewal and differentiation to enhance host fertility" was peer-reviewed at PLOS Biology. I apologize for the protracted review process for your study - in this case, for some reason, we had a bit of trouble lining up 3 reviewers in the initial stage and then I was at a conference all last week which delayed things further. Your manuscript has now been evaluated by the PLOS Biology editors, an Academic Editor with relevant expertise, and by several independent reviewers. 

In light of the reviews, which you will find at the end of this email, we would like to invite you to revise the work to thoroughly address the reviewers' reports.

As you will see below, the reviewers find the work of potential interest and the analyses to be fairly comprehensive. However they have raised a number of important and overlapping concerns that would need to be thoroughly addressed before we can consider your manuscript for publication. The reviewers have highlighted that the writing and data presentation is a bit challenging to follow, noting, for example that the discussion and connection of the discussion to figures is loose, and that some statements do not seem to be aligned with relevant figures or statistics. Additionally, the authors raise concerns about the appropriateness of the control genotypes used in several experiments and the reviewers have noted that comparisons of infected and uninfected wt flies are needed as a contrast to what happens in the mutants. After discussion with the Academic Editor we think these trials should be done in parallel, which might require a repeat of the whole experiment on fertility effects. 

Given the extent of revision needed, we cannot make a decision about publication until we have seen the revised manuscript and your response to the reviewers' comments. Your revised manuscript is likely to be sent for further evaluation by all or a subset of the reviewers.

**IMPORTANT - SUBMITTING YOUR REVISION**

*Re-submission Checklist*

*Published Peer Review*

*PLOS Data Policy*

*Blot and Gel Data Policy*

Sincerely,

Lucas

Lucas Smith, Ph.D.

Associate Editor

PLOS Biology

lsmith@plos.org

REVIEWS:

Reviewer #1: 

Since the early discovery that Wolbachia could rescue the sterility of some mutant alleles of the Sxl gene, additional studies have established Wolbachia as an important regulator of GSC maintenance and proliferation in the female germline. In this work, Russell et al. have now discovered that wMel is capable of rescuing several aspects of the complex oogenesis phenotype associated with mei-P26 knock-down or mutants in D. melanogaster. They performed a detailed analysis of these defects affecting GSC maintenance, germ cell proliferation and differentiation, as well as female fecundity and fertility. They also show that wMel is beneficial to the fertility of wild-type flies. One main interest of this work is to reveal the complexity of this critical host-symbiont interaction, with Wolbachia likely controlling several aspects of oogenesis through the expression of multiple proteins. 

- Having confirmed that CI is very weak in D. mel, the authors speculate on the possibility that wMel is losing its ability to induce CI and instead develops a mutualistic interaction with its host based on increased female fertility. However, the authors should explain how they integrate in their model the fact that wMel is fully capable of inducing strong CI in D. simulans or Aedes mosquitoes. 

- I found quite surprising that experiments in Figure 1A-C do not include a wild-type control. The authors refers to "wild-type" (l. 127) but it is actually missing in the Figure.

- The text alludes on several occasions on the "nature" of the mei-P26[1] allele (eg l.220, l.352). However, the mei-P26 alleles used in this work are not described. mei-P26[1] actually contains a transposon insertion in the first intron of the gene, so the actual impact of this mutation on the production of mei-P26 product is unclear. The authors could measure the amount of transcripts produced by this allele and the RNAi, compared to a control. 

- It was previously reported in Starr & Cline Nature 2002 that Wolbachia is incapable of rescuing the sterility of homozygous mei-P26[fs1] females. The fact that a complete null of mei-P26 is not rescuable should be clearly mentioned in the text and discussed. Similarly, l. 120, "flies lacking Mei-P26" is likely an overstatement. 

Additional comments:

- I am not sure if male-killing should be considered as a reproductive parasitism (l. 63).

- l. 70 "pipiensis" 

- l.143-144: I had hard time to understand this Table E until I read l. 163

- Fig 1J shows a perfect ovariole, yet fecundity is severely affected. A more global view of the ovary with several ovarioles could perhaps better reflect the reality.

- L. 201: the Wolbachia titers in somatic vs germ cells cannot really be appreciated in this image.

- L. 202: (Fig 1L-L') 

- L. 210: A description of GSC and other germ cells could help.

- L. 216: one extra "uninfected" word in the sentence.

- L. 230: I guess "PHH3" refers to phosphorylation of H3 serine 10.

- L. 242: Not sure to understand what the authors mean by "to recover some of their GSCs" or l. 245 "increased their GSCs abundance". 

- L. 288: This diagram is beautiful but unfortunately wrong. It is known since King (1970) that the oocyte invariably forms from one of the two cells with four ring canals.

- L. 375: "this dominant-negative …". I don't see any obvious dominant-negative effect here. This should be clarified.

- L.404-405: the use of the term "wild-type" is confusing in this sentence.

- L. 416: a color code is missing for Fig. 6B

- L. 557: This genotype is wrong.

Reviewer #2: In this manuscript, Russell et al. demonstrate that Wolbachia infection in Drosophila melanogaster is capable of rescuing defects in germline stem cell self-renewal and differentiation. Wolbachia bacteria of arthropods are some of the world's most successful endosymbionts in terms of spread to new hosts. Although the host reproductive manipulations of Wolbachia are infamously tied to their global spread, there is doubt that these manipulations alone could fully account for their success and many have speculated or investigated other contributions of Wolbachia to host fitness. The authors show that compensation for host reproductive defects may be one such mechanism, which is an important finding for basic and applied Wolbachia research. In particular, the authors identify mei-P26 as a specific host factor where a loss of function causes various developmental defects in oocytes that are rescued by Wolbachia infection. The study is comprehensive in its analysis and presents ample evidence of an interaction between the symbiont and mei-P26. The results are important for answering key questions in the field. There are some comments and suggestions below:

Comments:

* It would help to explain more in the main text how and why mei-P26 was chosen as a candidate. There is a section in the methods on it, but it would help at the beginning of the results or somewhere in the introduction to go into more detail on this. Since the paper focuses so much on it, it would help the reader to walk through the logic of how this candidate was identified out of all other possibilities. 

* How common are mei-P26 or similar GSC deficiencies in the wild? Would Wolbachia rescue of these defects be expected regularly, and would this ability commonly increase Wolbachia's fitness? 

* Would you expect to see mei-P26 mutants maintained in the wild with Wolbachia infection? 

* Some of the main comparisons of fertility across some fly genotypes are not quite appropriate. For example, the percentages highlighted in L124-130 compare the fertility of mei-P26 lines to wild-type lines. If the question being asked is whether Wolbachia rescues a mei-P26 deficit, then the main comparison should be between a mei-P26 deficient background either with or without the symbiont. It is ok to compare to the wild-type in other cases (for example, where the question being asked is how similar the Wolbachia rescue of defects is to wild-type). However, in the experiments with fertility, it would be most appropriate to focus primarily on within-genotype comparisons.

* Related to this, was there a control for the RNAi knockdown line that was used throughout the paper? I did not see one in the supplemental tables. For example, were RNAi knockdown results of mei-P26 ever compared to knockdown of a control gene like GFP? This is an important control to determine if there is an effect of the RNAi expression system on the phenotype. The inclusion of other mei-P26 deficient fly lines mitigates the concerns of the lack of this control somewhat, however, it may be beneficial (though perhaps not strictly necessary) to conduct one of the key assays with an RNAi control if there wasn't one already.

* It is somewhat difficult to follow the supplemental tables without spending significant time looking at them. This is primarily because there are many features that are not explained in the captions. Though many can be inferred by spending some time studying the tables closely, it would help readers to include more detail in the captions to avoid having to do this. For example, what do the colors mean? Why were the n values so different across comparison groups? What are the unfilled boxes? What are all of the shorthand notations for genotypes? Etc. It would help the readability of the tables to include more details on the meaning of each component.

* It would help to include a few more details on experimental design for the fertility crosses. Since these crosses are all being compared to each other, were they all conducted at the same time using similar controls and conditions? How many separate times were the experiments conducted/repeated? This is particularly important for the CI/rescue crosses, where they need to be performed on the same day for accurate comparison and other details are important to know: were the older and younger males all the first-emerged flies from their vials, did they all have similar-aged grandmothers, etc. It would help to have more of these details in the methods.

* There are some small details that would be good to include in figure captions that are missing. For example, what does each dot represent in the beeswarm boxplots? Are those hatch rates of individual females? Is this the combined data of multiple replicates? 

* For the fertility assays as well, a couple of minor points are unclear. It would help to explain the term "allelic strengths" (L123) the first time it is brought up, as it is unclear until later in the paragraph. 

* It may also help at the beginning of the results section to simply introduce the basic experimental setup, including which fly genotypes are being used and why, before going into the results. As is, the reader goes straight into the conclusion based on the results without being introduced to the assay that was performed.

* It is a little unclear what is meant by offspring production in Figure 1/S2/S5. Are the offspring produced not the same as the number of eggs laid or hatched?

* L155-156: It may help to add that FtsZ is a Wolbachia marker

* L179-181: It's stated here that wMel also rescues fertility impacts of mei-P26 in males, but the figure only shows a couple of significant differences, and in different genetic backgrounds across S2D-F, so it looks like the phenotype may not be as robust in males.

* FigS2G,H: It would help to have a few details on these vials, maybe in the methods or in the caption. Were these vials set up on similar days with similar numbers/ages of flies? It may help to label exactly what the most important differences are between the vials, particularly for people without fly experience. Is it just the color? If so, it may help to point out the color difference in the caption so readers know what to compare between the images. 

* L272/Figure 3: It doesn't look like region 2a is significant between the two mei-P26 groups

* Figure 6: It would help to add details on components of the figure like color and shape meanings.

Minor comments:

* There are some typos and small formatting errors in the document. For example, species names are not always italicized and the w in wMel should be italicized as well

* "pipientis" is misspelled in the introduction (L70)

* Some of the references to figures or tables are incorrect. These should be doublechecked throughout the document, for example:

o L137: should say Fig S2E-G, not S3E-G

o L141: is S3 correct here?

o L179: missing table reference?

o L181: should be Table S3 not S4

* Table S1/S2 titles: should "females" be plural? 

* The sentences in L168-171 seem to potentially be duplicates

Reviewer #3: Russell et al. have investigated how Wolbachia bacteria affect Drosophila fertility. They make a major discovery in finding that Wolbachia infection substantially rescues infertility caused by the mei-P26 mutation, and provide insight relating this to previous interactions observed with other germline genes. The data largely support the authors' conclusions (excepting point 4 below) but there are substantial issues with data presentation and organization that made this manuscript challenging to evaluate.

1) Results are often not connected to the relevant figure/table, or cite an incorrect table/figure. A wide range of figures is often cited at beginning of paragraphs with specific results that follow being not cited, leaving the reader to wade through multiple figures and tables to find the data. For examples, lines 124-130 appear to match to Fig 1E, but this is not stated. Lines 140-141 appear to match to Fig 1D but is cited as S3. Line 160 'differentiation defects' appears to refer to Fig 1H-K but is not cited. Data for the results in the paragraph starting on L160 are difficult to find. L244-246 - where is data?

2) Data presentation is confusing. Line 160/161 is redundant with L120/121. L161-164 is redundant with L124-125. L168-171 says the same thing twice.

There is confusion in some places whether the result being discussed is of mutant relative to wild type, or mutant with and without Wolbachia infection. For example, L166-167 appears to be making a statement about RNAi knockdown phenotype of mei-P26. But following sentence on Line 168-169 instead provides data on RNAi with and without infection.

The authors sometime start a paragraph stating a result, then give background/motivation on why doing the experiment, and then give more results. Which is confusing. Such as paragraphs on p. 15.

What's the difference between Fig 1G and S2A? 

L378. What are the 'differences among these tumor phenotypes'?

L178 - "across the fly lifespan". Does not seem to match Fig S2A-C, which shows effects reversing around day 30.

L214-214. Unclear what 'young' means here.

L434. 'differences … are insignificant'. In Table S4, some results are significant and some are not.

LL547. FS2 should be FS1?

Fig 1. Should emphasize that all data is from days 2-3 only (I think). 

Fig6A - what is parent age? 

Fig 6B,C - presumably is also OR-R?

3) Comparisons of mutants vs wild type. 

At many places mutant genotypes are compared for fertility phenotypes to wild type. One potential issue is experimental design. Presumably with and without Wolbachia comparisons within a genotype were done in parallel (at same time). Is that also true for different genotypes (e.g. OR-R and mei-P26 mutants)? If not, then that weakens the comparative power among genotypes. A second issue is the different genetic backgrounds. Different wild type backgrounds can have wide variation in fertility, and that will also occur here since mutants and wild type are different backgrounds. It therefore is unclear what the biological meaning is that some mutant conditions have higher fertility than wild-type (eg. L163-164). For these reasons, I would suggest de-emphasizing comparisons of fertility between mutants and wild-type, and focusing on paired comparisons of fertility phenotypes with and without Wolbachia within individual genotypes.

4) Wolbachia effects in wild type. 

I wouldn't call Gal4-expressing lines "wild type". At some places the authors seem to agree (lines 403-405), but at other places including in the Highlights and Abstract, they conflate the genotypes. This is important to clear up because a major claim of the paper is that Wolbachia increase fitness even in wild type individuals. If I understand all the data on this point, in true wild type (OR-R), only egg hatch rate is significantly increased by Wolbachia infection (S5C). But egg lay is somewhat reduced (S5B). Thus, total offspring is not different (S5A), which is the phenotype that ultimately matters for fitness. So the general conclusion that fitness is increased by infection in wild type lines does not seem to be established.

Minor points.

L395. Not sure 'recapitulates' is correct word here.

L462-3. Better to say 'rescue the PARTIAL loss'. The description of Sxl and Bam with the 'respectively' is confusing.

L585-590. How were long-term experiments with collections over 60 days done?

---

## [Decision Letter · Decision Letter 2]

16 Aug 2023

Dear Dr Russell,

Thank you for your patience while we considered your revised manuscript "Wolbachia endosymbionts manipulate GSC self-renewal and differentiation to reinforce host fertility" for publication as a Research Article at PLOS Biology. This revised version of your manuscript has been evaluated by the PLOS Biology editors, the Academic Editor and the original reviewers.

The reviewers are largely satisfied by the revision and have all suggested that we accept your study. However I note they raise a number of minor suggestions that we think should be addressed before publication. Additionally, before we can accept your study, we need you to address a number of editorial requests detailed below. We therefore would like to invite one last revision that we do not think will take very long. 

**EDITORIAL REQUESTS: 

1) TITLE: We suggest that the title be edited to spell out GSC and specify the host, as this will make the study more broadly accessible. If you agree, we suggest you change the title to something like: 

"Wolbachia endosymbionts manipulate the self-renewal and differentiation of germline stem cells to reinforce fertility of their fruit fly host"

2) ABSTRACT: Similarly, GSC should be defined in the abstract

3) FINANCIAL DISCLOSURES: Please update the financial disclosures statement, in our editorial manager system, to describe the role of any sponsors or funders in the study design, data collection and analysis, decision to publish, or preparation of the manuscript. If the funders had no role in any of the above, include this sentence at the end of your statement: "The funders had no role in study design, data collection and analysis, decision to publish, or preparation of the manuscript."

4) DATA: Thank you for providing data on NCBI and Dryad related to your manuscript. For some reason, I could not access these datasets to verify that they meet our requirements. Can you provide me with a reviewer code? (sorry if I missed this somewhere)

>>Please do take a look at our Data Policy, which requires that all data be made available without restriction, and make sure that your data provided on these repositories meets our requirements. Note that we do not require all raw data. Rather, we ask that all individual quantitative observations that underlie the data summarized in the figures and results of your paper be made available either as a deposition on a repository or as a supplementary excel file. 

>>Please also ensure that figure legends in your manuscript include information on where the underlying data can be found. For example, to each figure legend you can add the sentence "the data underlying this figure can be found at ____"

We expect to receive your revised manuscript within two weeks. 

*Published Peer Review History*

*Press*

Sincerely,

Luke

Lucas Smith, Ph.D.

Senior Editor,

lsmith@plos.org,

PLOS Biology

Reviewer remarks:

Reviewer #1: My main requests have been addressed. As for my minor comment on "other germ cells", I meant nurse cells and the oocyte. 

Reviewer #2: The authors have thoroughly addressed comments by the reviewers. In particular, they have made many clarifying statements regarding the methods, addressed concerns with figure/table citations in the text, and addressed concerns with the fertility assays. The new details on the methods for the fertility assays are helpful. The authors have also added comparisons of fertility within genotype which, paired with the comparisons of fertility to controls (and their justification for this is good), establish their points well. This addresses the questions raised by reviewers. I have only a couple of very minor suggestions remaining:

I suggest a few very minor clarifying details on the caption of Figure S4H,I (the stock images): 1) I would label the dates that each individual vial received adults (presumably older on the left to newer on the right of each image?). This would help readers know which vials should be compared to which other vials between images. 2) I would also suggest pointing out the mold growth is visible by different food colors/consistencies between comparably-aged vials in H and I. 3) I would clarify that the sentence starting with "Infection enables stable robust stock persistence..." is referring to the Wolbachia infection as opposed to the mold.

Also, very minor: if the authors have agreed to go with the convention of italicizing the w in wMel, then this should be also applied to supplemental documents (supplemental figure captions etc.)

Reviewer #3: The authors have clarified many of the concerns and queries about the fertility assays. They have also added extensive discussion of new RNA-seq experiments. What does "dual-RNAseq" mean - examining both Dmel and wMel genes, or combining Or-R and mei-P26 in a single analysis? 

I didn't see the following in Fig 1, referring to the time frame of the eggs/offspring being analyzed.

Fig 1. Should emphasize that all data is from days 2-3 only (I think). 

Thank you for pointing this out. We added this info to the figure legend to make sure it is clear.

---

## [Editor Report · Decision Letter 3]

14 Sep 2023

Dear Dr Russell,

Thank you for the submission of your revised Research Article "Wolbachia endosymbionts manipulate the self-renewal and differentiation of germline stem cells to reinforce fertility of their fruit fly host" for publication in PLOS Biology, and thank you for addressing the last reviewer and editorial requests in this revision. On behalf of my colleagues and the Academic Editor, Nancy A. Moran, I am pleased to say that we can in principle accept your manuscript for publication, provided you address any remaining formatting and reporting issues. These will be detailed in an email you should receive within 2-3 business days from our colleagues in the journal operations team; no action is required from you until then. Please note that we will not be able to formally accept your manuscript and schedule it for publication until you have completed any requested changes.

**IMPORTANT: As you address any formatting and reporting requests, to come, please also address the following editorial points which we discussed over email: 

1) Please update the data availability statement in our editorial manager system, and the relevant sections of the manuscript, to reference the correct BioProject number PRJNA1007602 

2) Please update the supplemental figure legends with a short sentence pointing readers to the relevant underlying datasets. 

PRESS

Sincerely, 

Luke

Lucas Smith, Ph.D.,

Senior Editor

PLOS Biology

lsmith@plos.org